# Discovery of a Drug-like, Natural Product-Inspired DCAF11 Ligand Chemotype

Gang Xue[1,4], Jianing Xie[1,4], Matthias Hinterndorfer[2], Marko Cigler[2], Lara Dötsch[1,3], Hana Imrichova[2], Philipp Lampe[1], Xiufen Cheng[1], Soheila Rezaei Adariani[1], Georg E. Winter[2]✉ & Herbert Waldmann[1,3]✉

Targeted proteasomal and autophagic protein degradation, often employing bifunctional modalities, is a new paradigm for modulation of protein function. In an attempt to explore protein degradation by means of autophagy we combine arylidene-indolinones reported to bind the autophagy-related LC3B-protein and ligands of the PDEδ lipoprotein chaperone, the BRD2/3/4-bromodomain containing proteins and the BTK- and BLK kinases. Unexpectedly, the resulting bifunctional degraders do not induce protein degradation by means of macroautophagy, but instead direct their targets to the ubiquitin-proteasome system. Target and mechanism identification reveal that the arylidene-indolinones covalently bind DCAF11, a substrate receptor in the CUL4A/B-RBX1-DDB1-DCAF11 E3 ligase. The tempered α, β-unsaturated indolinone electrophiles define a drug-like DCAF11-ligand class that enables exploration of this E3 ligase in chemical biology and medicinal chemistry programs. The arylidene-indolinone scaffold frequently occurs in natural products which raises the question whether E3 ligand classes can be found more widely among natural products and related compounds.

Targeted protein degradation (TPD) has emerged as a new paradigm for modulation of protein activity in chemical biology and drug discovery[1-3]. A viable strategy for TPD employs bifunctional protein targeting chimeras (PROTACs), in which small molecule protein ligands are linked to a ligand of an E3 ligase to yield bifunctional modalities that induce degradation of the protein of interest by the ubiquitin-proteasome system (UPS)[4,5]. Recently also bifunctional degrader modalities have been reported that direct the protein of interest to the autophagy-related proteins LC3B or p62, thereby inducing protein degradation by means of macroautophagy[6,7]. In particular, it was reported that α, β-unsaturated indolinone **1** (R = I; GW5074, Fig. 1a) targets LC3B and thereby induces degradation of huntingtin by macroautophagy[8], and that a bifunctional degrader derived from **1** primes the BRD4 protein for autophagic degradation[6].

PROTAC-mediated degradation through the UPS has been applied to the lipoprotein-binding chaperone PDEδ[9] which regulates the cellular distribution, and thereby the activity of the Ras-oncoproteins and other lipidated proteins. However, cellular ligand binding to PDEδ is antagonized by Arl2/3-mediated cargo release[10,11], such that full proteasomal degradation of the chaperone could not be achieved. Therefore, we set out to explore whether effective degradation of PDEδ would be possible by means of macroautophagy, initiated by a bifunctional modality combining ligands of LC3B and PDEδ.

We describe the design and application of bifunctional degraders employing α, β-unsaturated lactams **1** (R = I) and **2** (R = Br) to direct proteins of interest into the cellular degradation machinery (Fig. 1a). We report that, unexpectedly, degradation mediated by bifunctional modalities incorporating **1** or **2** does not proceed via the autophagy pathway but rather by means of the UPS. We identify the CUL4A/B-RBX1-DDB1-DCAF11 (CRL4^DCAF11) complex as E3 ligase recruited by the bifunctional degraders to the proteins of interest for initiation of degradation, and show that the substrate receptor

[1]Department of Chemical Biology, Max Planck Institute of Molecular Physiology, Dortmund, Germany. [2]CeMM Research Center for Molecular Medicine of the Austrian Academy of Sciences, Vienna, Austria. [3]Technical University Dortmund, Faculty of Chemistry and Chemical Biology, Dortmund, Germany. [4]These authors contributed equally: Gang Xue, Jianing Xie. ✉e-mail: GWinter@cemm.oeaw.ac.at; herbert.waldmann@mpi-dortmund.mpg.de

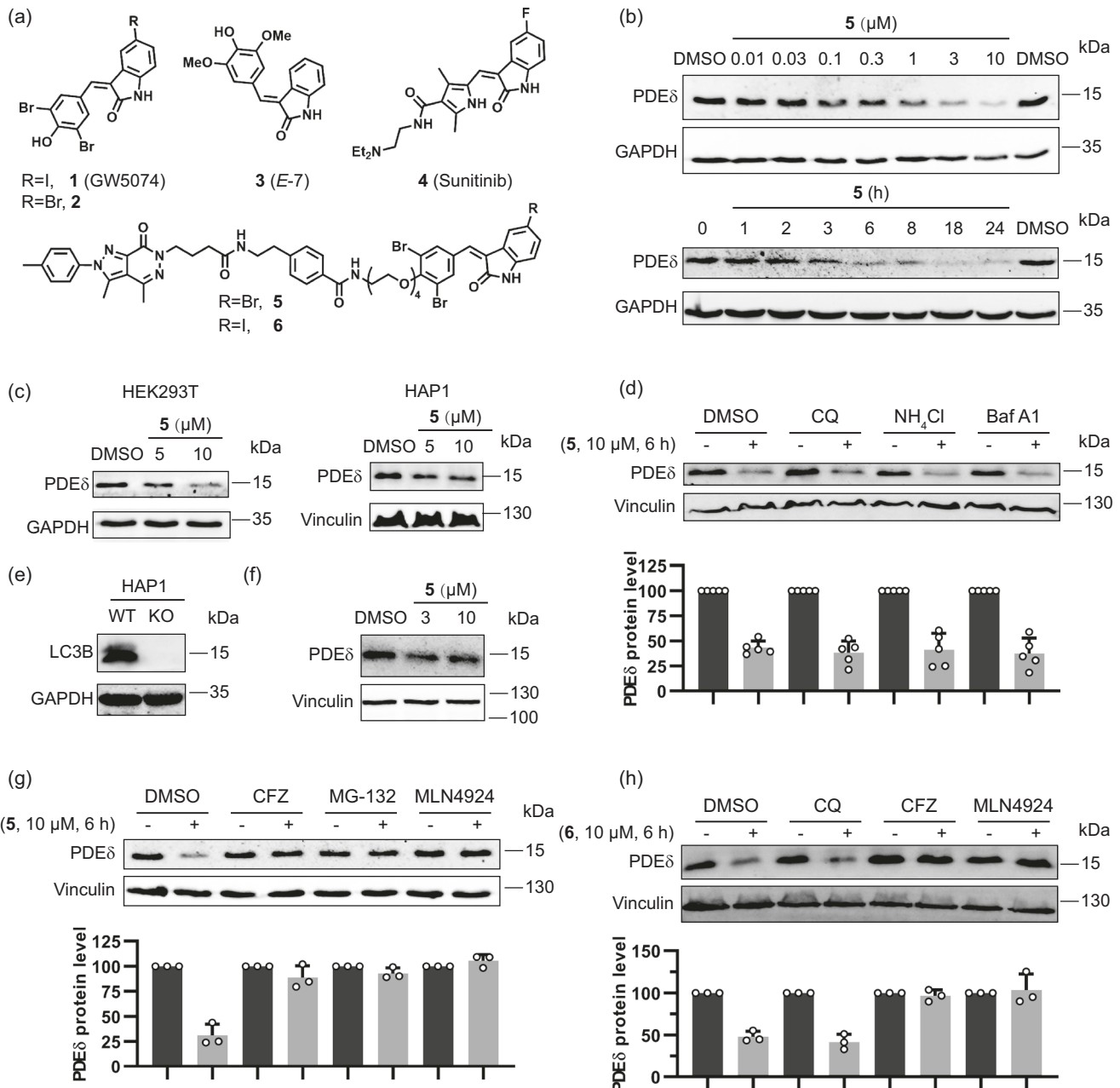

**Fig. 1 | Degradation of PDEδ induced by bifunctional arylidene-indolinones 5 and 6. a** Structures of selected indolinones and bifunctional degraders derived therefrom. **b** Dose-dependent degradation of PDEδ in Jurkat cells treated with compound **5** (6 h) and time-dependent degradation of PDEδ in Jurkat cells treated with compound **5** (10 μM). Representative result of $n = 3$. **c** PDEδ levels in HEK293T and HAP1 cells treated with compound **5** at 5 μM or 10 μM. Representative result of $n = 3$. **d** PDEδ levels in Jurkat cells pretreated with autophagy inhibitors Chloroquine (CQ, 50 μM), NH$_4$Cl (15 mM), and Bafilomycin A1 (Baf A1, 0.3 μM) for 2 h prior to the addition of 10 μM compound **5** for another 6 h. Quantification of the relative PDEδ protein content in relation to the DMSO control is shown in the bar graph. Data are mean values ± SD ($n = 5$ biological replicates). **e**, **f** PDEδ levels in LC3B knockout HAP1 cells. Representative result of $n = 3$. **e** Lack of LC3B protein in LC3B knockout

HAP1 cells. **f** LC3B knockout HAP1 cells were treated with compound **5**, and PDEδ levels were analyzed using immunoblotting. **g** PDEδ levels in Jurkat cells pretreated with the proteasome inhibitors Carfilzomib (CFZ, 0.2 μM) or MG-132 (10 μM), or the neddylation inhibitor MLN4924 (1 μM) for 2 h, followed by treatment with 10 μM of compound **5** for another 6 h. Quantification of the relative PDEδ protein content in relation to the DMSO control is shown in the bar graph. Data are mean values ± SD ($n = 3$ biological replicates). **h** Western blots of PDEδ in Jurkat cells pretreated with CQ (50 μM), CFZ (0. 2 μM) or MLN4924 (1 μM) for 2 h, followed by treatment with 10 μM compound **6** for another 6 h. Quantification of the relative PDEδ protein content in relation to the DMSO control is shown in the bar graph. Data are mean values ± SD ($n = 3$ biological replicates).

DCAF11 is covalently targeted by the 3-arylidene-indolinone moiety. This α, β-unsaturated cyclic amide scaffold occurs in various bioactive natural products, e.g. the alkaloid *E*-7 (**3**, Fig. 1a)[12] and is drug-like. For instance, compound **1** (GW5074) has been explored in kinase- and SIRT-inhibitor discovery[13–15] and is an established research tool[16]. It shows a good safety profile[17] and can be considered a close analog of the marketed kinase inhibitor sunitinib (**4**,

Fig. 1a). DCAF11 has only been targeted by chloroacetamide derivatives before, which were considered valuable tool compounds for cellular investigations[18], and the discovery of degraders incorporating more tempered electrophiles, such as α, β-unsaturated amides was deemed highly desirable. The drug-like, natural-product inspired arylidene-indolinones define a DCAF11-ligand class with balanced electrophilic reactivity that may enable the wider

exploration of this E3 ligase in chemical biology and medicinal chemistry programs.

## Results

### PDEδ is degraded through the UPS, not through macroautophagy

For the design of bifunctional degraders, we drew from the structure-activity correlations previously established for PDEδ ligands deltazinone[19] and deltasonamide[11] and PDEδ PROTACs based on these two PDEδ ligands[9,20] (Supplementary Fig. 1a). Protein-ligand complex crystal structures[11] and a docking model[19] had informed probe- and PROTAC development and shown that linkers are best attached through benzamides, extending out of the PDEδ binding site leading to bifunctional compounds **5, 6** and **7** (Fig. 1a, and Supplementary Fig. 1a, b).

GW5074 had been identified as LC3B binder after immobilization on a solid support through the phenolic OH-group, suggesting that a linker could be attached at this site[8]. In the discovery of GW5074 both an iodide and a bromide were identified as suitable substituents in the 5-position of the indolinone[14] and we explored both substituents (Fig. 1a, compounds **5** and **6**).

Treatment of Jurkat cells with bromine-substituted deltazinone-derived probe **5** reduces PDEδ levels in dose- and time-dependent manner (Fig. 1b). Up to 10 μM concentration of **5**, no hook effect, which often is characteristic for bifunctional degraders, was observed[21]. At 10 μM degrader **5** induced ca. 75 % degradation after 18 h incubation, and strong degradation was apparent already at 3 h. PDEδ was also degraded by **5** in HEK293T and HAP1 cell lines (Fig. 1c). Iodine-derivative **6** and deltasonamide-derived compound **7** also induced degradation of PDEδ with similar activity as displayed by compound **5** (Supplementary Fig. 1 c–e).

PDEδ degradation occurred post-transcriptionally since no reduction in mRNA levels was observed in Jurkat cells after 6 h treatment with **5** (Supplementary Fig. 1f). Deltazinone or **2** alone, or in combination did not reduce PDEδ protein levels (Supplementary Fig. 1g). In a recent report, a bifunctional degrader incorporating indolinone GW5074 was described which targets BET proteins for autophagic degradation[6]. Hence, we expected that PDEδ degradation induced by the bifunctional ligands described above would also involve macroautophagy as cellular degradation mechanism. However, after pretreatment of Jurkat cells with the autophagy inhibitors Chloroquine (CQ), $NH_4Cl$ or Bafilomycin A1 (Baf A1) for 2 h, treatment with 10 μM bifunctional deltazinone-derivative **5** for 6 h still led to degradation of PDEδ, indicating that macroautophagy at best only plays a minor role in the degradation process (Fig. 1d). In addition, compound **5** also induced PDEδ degradation in LC3B knockout HAP1 cells (Fig. 1e, f) demonstrating that LC3B is not involved in the degradation mechanism. In contrast, after pre-treatment of Jurkat cells with the proteasome inhibitors Carfilzomib (CFZ) and MG-132, degradation of PDEδ by bifunctional compound **5** was abrogated, suggesting that the proteasome might be involved in the degradation (Fig. 1g). Proteasomal degradation mediated by Cullin-Ring Ubiquitin ligases (CRLs) is dependent on modification with NEDD8 on the cullin backbone. In the presence of the neddylation inhibitor MLN4924[22], degradation of PDEδ did not occur (Fig. 1g). Similar results were obtained when degrader **6** with GW5074 as a LC3B ligand and deltasonamide-derived bifunctional degrader **7** were employed (Fig. 1h and Supplementary Fig. 1h). These findings indicate that degradation of PDEδ induced by bifunctional modalities **5, 6** and **7** does not occur by means of macroautophagy but rather proceeds through the UPS and is mediated by a CRL E3 ligase.

### BET proteins and kinases are degraded through the UPS

In order to explore whether other proteins could be subjected to degradation by bifunctional compounds incorporating arylidene-indolinones, we combined the BET protein ligand JQ1 **8** with bromine-substituted indolinone **2**. JQ1 has been used repeatedly for PROTAC development and a suitable site for linker attachment is established (Fig. 2a)[23]. Ligands **2** and **8** were connected by a variety of different linkers (Supplementary Fig. 2a). All bifunctional probes induced degradation of BET proteins BRD2, 3 and 4 in Jurkat cells at 3 μM after 6 h treatment. Compounds **9** and **10** showed potent degradation activity, and were chosen for subsequent experiments (Fig. 2b and Supplementary Fig. 2b).

BRD2, 3 and 4 degradation by **9** is concentration- and time-dependent in Jurkat cells (Fig. 2b, c) and BRD2 depletion was also observed in Hep G2 and MDA-MB-231 cells (Supplementary Fig. 2c). Compound **9** induced degradation of BRD2 with a maximum level of degradation ($D_{max}$) of 84 % at 3 μM and a half-maximal degradation concentration ($DC_{50}$) of 255 nM (Fig. 2b and Supplementary Fig. 2d). Up to 50 μM concentration, almost no hook effect was observed for BRD2 and BRD4, but at 10 μM, a strong hook effect was observed for BRD3 (Fig. 2b). BET mRNA levels were not reduced in Jurkat cells after treatment with **9** (Supplementary Fig. 2e). JQ1 (**8**) or **2** alone or in combination did not reduce BRD2 protein levels (Supplementary Fig. 2f). Compounds **9** and **10** carry a bromine-substituent in the 5-position of the indolinone, whereas in GW5074 an iodine occupies this position. To determine whether the type of halogen influences BET-protein degradation, we treated Jurkat cells with compound **10** and its iodo analog **11** and observed comparable efficiency of BRD2 degradation (Supplementary Fig. 2g). Wang and Ouyang et al. recently reported that close analogs of **11** induce degradation of BRD4 through autophagy (Supplementary Fig. 3a)[6], thus we tested the role of autophagy for BET protein degradation by compounds **9, 10** or **11**. Similar to the observations made for PDEδ degradation, treatment with the autophagy inhibitor CQ did not prevent BRD2 degradation, whereas in the presence of the proteasome inhibitor CFZ or the ned-dylation inhibitor MLN4924, no degradation was observed (Fig. 2d, Supplementary Fig. 3b–d). To further validate that macroautophagy is functionally not involved in the observed compound-induced target degradation, we generated two independent knockout pools of the essential autophagy factor FIP200 in KBM7 cells (Supplementary Fig. 3e). Supporting our hypothesis, BRD3 and BRD4 degradation was not influenced by FIP200 levels (Supplementary Fig. 3f). These findings suggest that also degradation of BET proteins proceeds via the UPS and not by autophagy.

As a further example for arylidene-indolinone-based bifunctional degraders, compound **13** that is based on ibrutinib (**12**) was explored (Fig. 2e). Ibrutinib inhibits BLK and BTK kinases with apparent $IC_{50}$ values of 0.5 nM[24]. Upon treatment of Ramos cells with degrader **13** for 6 h, BLK and BTK levels were reduced by 70% and 50%, respectively (Fig. 2f, g). Again, the proteasome inhibitor CFZ or neddylation inhibitor MLN4924 abolished degradation but treatment with the autophagy inhibitor CQ did not (Fig. 2h). These results demonstrate that bifunctional moieties composed of GW5074 (**1**) or its analog (**2**) and a suitable protein ligand induce degradation of different proteins through the UPS by recruiting a neddylation-dependent E3 ligase, presumably a CRL.

### CRL4$^{DCAF11}$ is recruited for proteasomal degradation of target proteins

To identify the E3 ligase targeted by the degraders, we performed a fluorescence-activated cell sorting (FACS)-based CRISPR/Cas9 screen (Fig. 3a). To this end, we engineered a fluorescent BRD4 protein stability reporter by fusing BRD4S to TagBFP, followed by a P2A self-cleaving peptide and mCherry for normalization. We stably expressed this reporter in KBM7 cells harboring a doxycycline-inducible Cas9 allele[25] and then transduced BRD4 reporter cells with a CRL-focused sgRNA library[26] targeting 495 genes with 8 sgRNAs per gene. After selection of library expressing cells, Cas9 expression was induced with

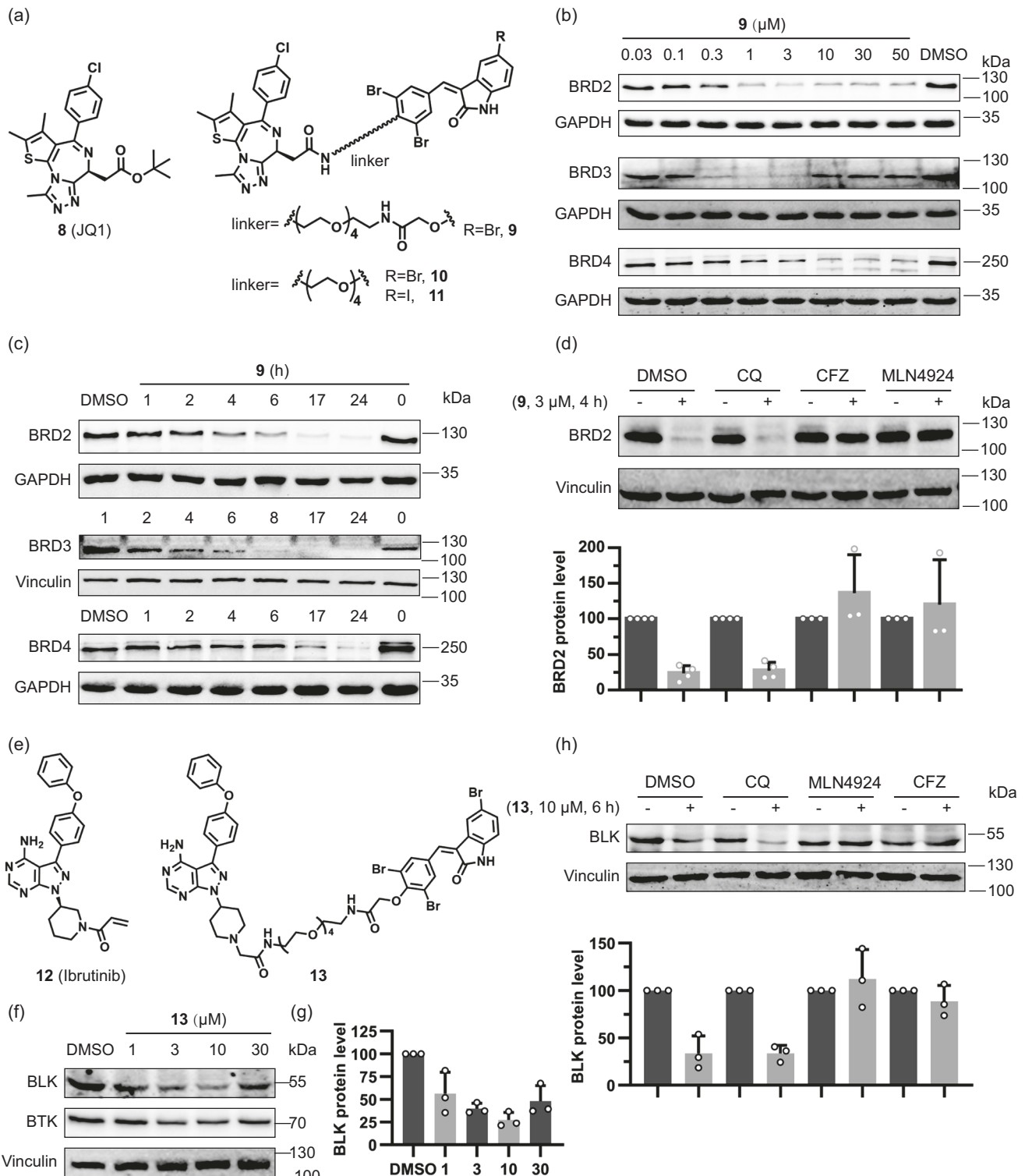

**Fig. 2 | Degradation activity and mechanism of action of bifunctional compounds that target BET proteins or BTK/BLK. a** Structures of the BET inhibitor JQ1 and selected bifunctional degraders. **b** Dose-dependent degradation of BET proteins in Jurkat cells treated with compound **9** for 6 h. Representative result of $n = 3$ is shown. **c** Time-dependent degradation of BET proteins in Jurkat cells treated with compound **9** at 3 μM. Representative result of $n = 3$. **d** BRD2 protein levels in Jurkat cells pretreated with the autophagy inhibitor CQ (50 μM), proteasome inhibitor CFZ (1 μM) and neddylation inhibitor MLN4924 (1 μM) for 40 min, followed by treatment with 3 μM compound **9** for another 4 h. Quantification of the relative BRD2 protein content in relation to the DMSO control is shown in the bar graph. Data are mean values ± SD ($n = 4$ biological replicates for DMSO and CQ;

$n = 3$ biological replicates for MLN4924 and CFZ). **e** Structures of the BTK- and BLK inhibitor ibrutinib and related bifunctional degrader. **f** Dose-dependent degradation of BTK- and BLK proteins in Jurkat cells treated with compound **13** for 6 h. Representative result of $n = 3$ is shown. **g** Quantification of the relative BLK protein content, which was normalized to the DMSO control. Data are mean values ± SD ($n = 3$ biological replicates). **h** BLK protein levels in Ramos cells pretreated with the autophagy inhibitor CQ (50 μM), proteasome inhibitor CFZ (1 μM) and neddylation inhibitor MLN4924 (1 μM) for 40 min, followed by treatment with 10 μM of compound **13** for another 6 h. Quantification of the relative BLK protein content in relation to the DMSO control is shown in the bar graph. Data are mean values ± SD ($n = 3$ biological replicates).

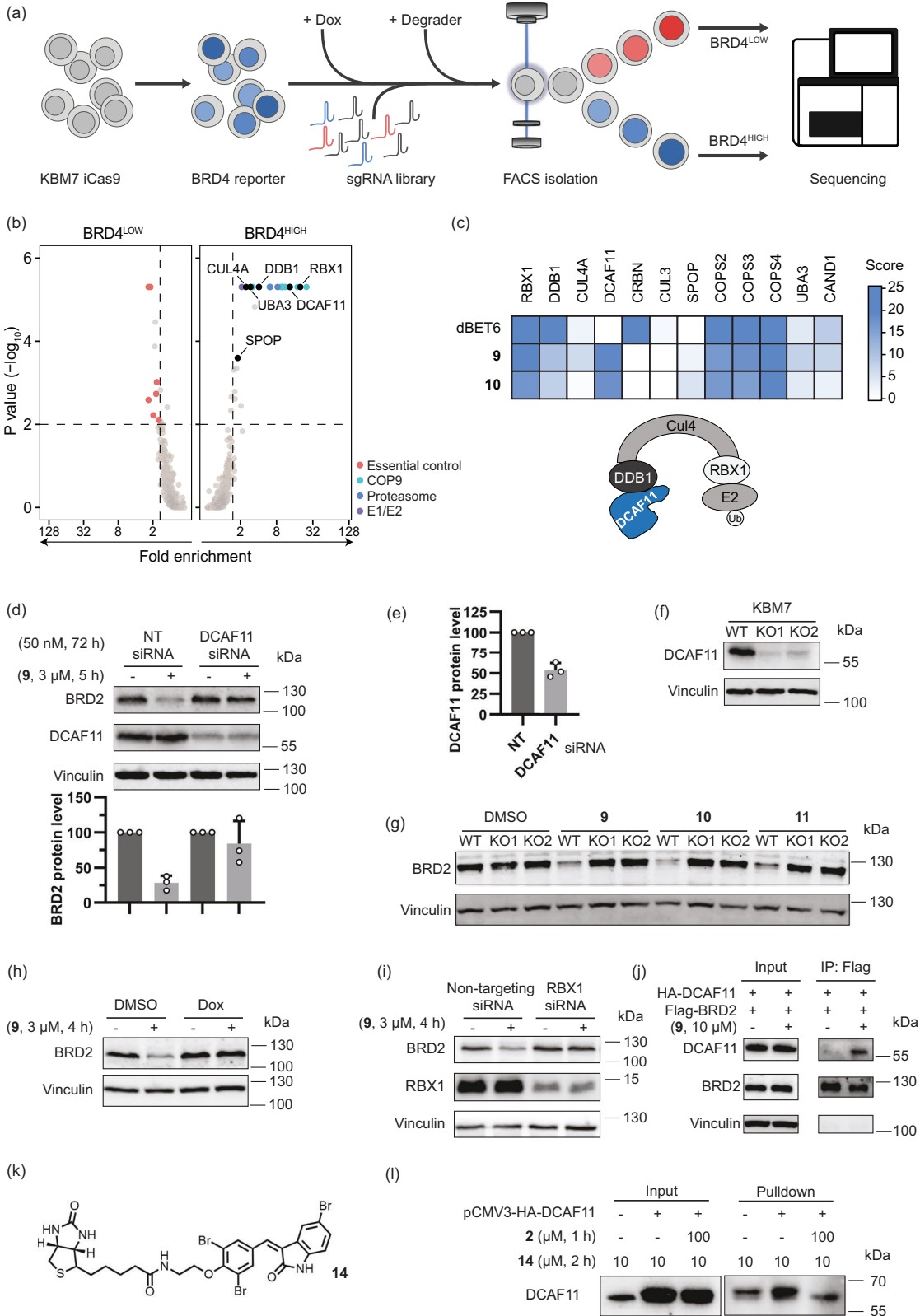

doxycycline for 72 h, followed by treatment with compounds **9** and **10** or the established BRD4 PROTAC dBET6[27]. Cells showing elevated (BRD4[HIGH]) or decreased (BRD4[LOW]) BFP levels compared to the bulk population (BRD4[MID]) were isolated via FACS and the sgRNAs enriched in the respective populations were quantified via deep-sequencing (Fig. 3a). As expected, the dBET6 control screen identified the members of the CUL4[CRBN] E3 ligase complex, including the scaffolding protein CUL4A, the E2-adaptor ring box protein RBX1, the DNA damage binding protein DDB1 and the substrate receptor CRBN as top hits (Supplementary Fig. 4a and Supplementary Data 1 and 2). Further hits included 20 S proteasome subunits and general regulators of CRL activity, such as the COP9 signalosome and the NEDD8-activating E1 enzyme subunit UBA3, thus confirming the successful identification of genes required for BRD4 degradation via dBET6 in the FACS-based

**Fig. 3 | Compound 9 promotes proteasomal degradation of BET proteins via the CRL^DCAF11 complex. a** Schematic of FACS-based BRD4 protein stability CRISPR screen assay. **b, c** Identification of genes required for compound 9-mediated degradation of BRD4. **b** FACS-based CRISPR screens for regulators of BRD4 degradation induced by 9. Average gene-level fold-changes and non-adjusted one-sided *p*-values of BRD4^HIGH and BRD4^LOW cell populations compared to BRD4^MID fraction were calculated using MAGeCK[46]. Essential control genes (BRD4^LOW) and 20S proteasome subunits, COP9 signalosome subunits and E1 or E2 ubiquitin ligases (BRD4^HIGH) inside the scoring window (*p*-value < 0.01, fold-change > 1.5) are labelled. **c** Heatmap of selected screen hits from Fig. 3b and Supplementary Fig. 4a, b. Gene-level score was calculated as Log₂ (fold change) * Log₁₀ (*p*-value). **d, e** BRD2 and DCAF11 protein levels in Hep G2 cells transfected with non-targeting (NT) siRNA or DCAF11 siRNA, followed by 9 addition. **d** Quantification of the relative BRD2 levels. Data are mean values ± SD (*n* = 3 biological replicates). **e** Quantification

of the relative DCAF11 levels. Data are mean values ± SD (*n* = 3 biological replicates). **f** DCAF11 levels in wild type (WT) KBM7 cells and DCAF11 knockout (KO1 and KO2) KBM7 cells. Representative result of *n* = 3. **g** BRD2 levels in WT and DCAF11 knockout (KO1 and KO2) KBM7 cells treated with compound 9 at 3 μM, 10 at 10 μM and 11 at 10 μM, for 6 h. Data are representative of *n* = 3. **h** BRD2 levels in Dox-inducible DDB1 knockout KBM7 cells pretreated with 0.5 μg/mL Dox for 3 days followed by addition of 9 at 3 μM for another 4 h. Representative result of *n* = 3. **i** BRD2 and RBX1 levels in MDA-MB-231 cells with NT or RBX1 siRNA followed by 9 addition. Representative result of *n* = 3. **j** BRD2 co-immunoprecipitation. FLAG-BRD2 protein was enriched by anti-FLAG beads and samples were analyzed for DCAF11. Representative result of *n* = 2. **k** Structure of biotinylated affinity probe 14. **l** Affinity-based enrichment by pulldown in Hep G2 cells overexpressing DCAF11. Protein levels were analyzed by WB. Representative result of *n* = 2.

CRISPR screen (Fig. 3c, Supplementary Fig. 4a and 5). For compounds 9 and 10, similar genes were discovered, however, DCAF11 rather than CRBN scored as the most significant CRL substrate receptor in both screens (Fig. 3b, c, Supplementary Fig. 4b and Supplementary Data 1 and 2). These data suggest that bifunctional compounds 9 and 10 recruit the CRL4^DCAF11 complex to induce proteasomal degradation of BRD4 (Fig. 3c).

For target validation, DCAF11 was depleted by RNA interference (Fig. 3d, e), which abolished BRD2 degradation induced by 9 (Fig. 3d). In addition, two different DCAF11-targeting sgRNAs were used to generate stable knockouts in KBM7 cells (KO1 and KO2, Fig. 3f). In both cases, depletion of DCAF11 abolished the BRD2 degradation induced by compounds 9, 10, and 11 (Fig. 3g). Similar results were obtained for BRD3 and BRD4 (Supplementary Fig. 4c) and PDEδ degradation by 5 (Supplementary Fig. 4d). Additionally, after knockout of the DDB1 adaptor protein by Dox-inducible CRISPR/Cas9 expression in KBM7 cells (Supplementary Fig. 4e) treatment with degraders 9 and 10 did not lead to BRD2 degradation anymore (Fig. 3h and Supplementary Fig. 4e), and degradation of PDEδ by treatment with 5 was also abolished (Supplementary Fig. 4f). After siRNA-mediated depletion of the RING box protein RBX1, compound 9 did not induce degradation of BRD2 anymore (Fig. 3i).

For confirmation of direct target engagement, FLAG-tagged BRD2 and HA-tagged DCAF11 were expressed in Hep G2 cells. Co-immunoprecipitation of FLAG-BRD2 enriched DCAF11 in the presence of bifunctional compound 9, indicating that 9 mediates complex formation between BRD2 and DCAF11 (Fig. 3j). To analyze if compound 2 binds to DCAF11 in cells, we connected biotin to compound 2 and synthesized probe 14 (Fig. 3k). After incubating 14 with Hep G2 cells harboring HA-DCAF11 overexpression for 2 h with or without compound 2 pretreatment, we performed affinity-based enrichment (biotin-streptavidin pulldown), and found that probe 14 enriched DCAF11 protein, while compound 2 reduced the enrichment (Fig. 3l), which indicated compound 2 binding to DCAF11 in cells.

**Arylidene-indolinones covalently target DCAF11**

DCAF11 exposes three reactive cysteines (C443, C460 and C485) in its putative small-molecule binding site which are targeted by the previously identified chloroacetamide 15[18]. Even though, to the best of our knowledge, covalent targeting by GW5074 or sunitinib has not been reported, α, β-unsaturated indolinones do embody a potential electrophilic Michael acceptor. We hence wanted to investigate a putative covalent binding to DCAF11. Indeed, while α,β-unsaturated degraders 9 and 10 displayed BRD2-degrading activity, the corresponding saturated analogs 16 and 17 did not induce degradation of BRD2 (Fig. 4a, c and Supplementary Fig. 6a).

MZ1 is a non-covalent, reversible VHL-based BRD2 degrader which can be washed out (Fig. 4d and Supplementary Fig. 6b)[28]. In contrast, in the presence of α, β-unsaturated indolinone 9 under the same washout conditions BRD2 levels did not recover (Fig. 4d). In addition, blocking

cysteine residues C443, C460 and C485 by treatment of Hep G2 cells with the aforementioned chloroacetamide 15, impaired the degradation of BRD2 by bifunctional degrader 9, thus implying that the α, β-unsaturated indolinones engage the same cysteine hotspot (Fig. 4e). Indeed, when DCAF11 knockout KBM7 cells were stably reconstituted with *DCAF11* cDNA carrying the respective triple mutation (C443A, C460A and C485A), degradation by 9 and 10 could not be observed. In contrast, reconstitution with *wildtype DCAF11* cDNA rescued degrader efficacy to levels comparable to unmodified KBM7 cells (Supplementary Fig. 4g). Of note, BRD4 degradation by the CRBN-dependent degrader dBET6 was not affected in any of the assayed conditions (Supplementary Fig. 4g), and protein abundance of *wildtype* and mutant DCAF11 were comparable (Supplementary Fig. 4h). Finally, direct binding to DCAF11 was proven by means of BODIPY-labelled analog 18 (Fig. 4b). This fluorescent compound binds to purified DCAF11 in a concentration-dependent manner, and binding is prevented by addition of non-labelled monovalent indolinone 2 (Fig. 4f and Supplementary Fig. 6c). In conclusion, the data obtained for fluorescent-labelled probe 18 and biotinylated probe 14 is consistent with direct binding through covalent attachment of the α, β-unsaturated compounds to DCAF11 in cells.

**Structure-activity relationship for DCAF11-targeting arylidene-indolinones**

For targeted protein degradation, currently mostly PROTACs based on the E3 ligases VHL and Cereblon[5] are employed. However, targeting of these ligases has been accompanied with occurrence of mutations and resistance[29–31]. In addition, different ligases can display different degradation efficiencies for a given target (see discussion in ref. 4). This can be associated with differential expression levels of a ligase in different tissues or cell types, with a particular subcellular localization, or with compatibility between the protein-protein interface formed between ligase and target. Hence, targeting of alternative E3 ligases for TPD may offer new opportunities and is highly desirable. For such ligases, modulation of activity should follow a structure-activity relationship to guide optimization of potency and selectivity. Initial investigations had revealed that variation of linker length clearly influences degradation efficiency of the DCAF11 E3 ligase complex (Supplementary Fig. 2a, b). For further improvement of DCAF11-ligand binding affinity, the substitution pattern on the indolinone scaffold was explored. To this end, we developed an In-Cell Western (ICW) assay[32] monitoring BRD2 levels in Hep G2 cells (Supplementary Fig. 7a). Treatment of Hep G2 cells with 9 for 4 h and subsequent analysis by In-Cell Western confirmed dose-dependent depletion of BRD2 by compound 9 (Fig. 4g), whereas saturated analog 16 did not induce *in cellulo* degradation (Supplementary Fig. 7b).

Competition with covalently labelling indolinones (e.g. 2) not equipped with the JQ1-ligand will occupy the DCAF11 binding site and, thereby, prevent BRD2 degradation such that higher BRD2 levels after pretreatment indicates more potent DCAF11-labelling by the

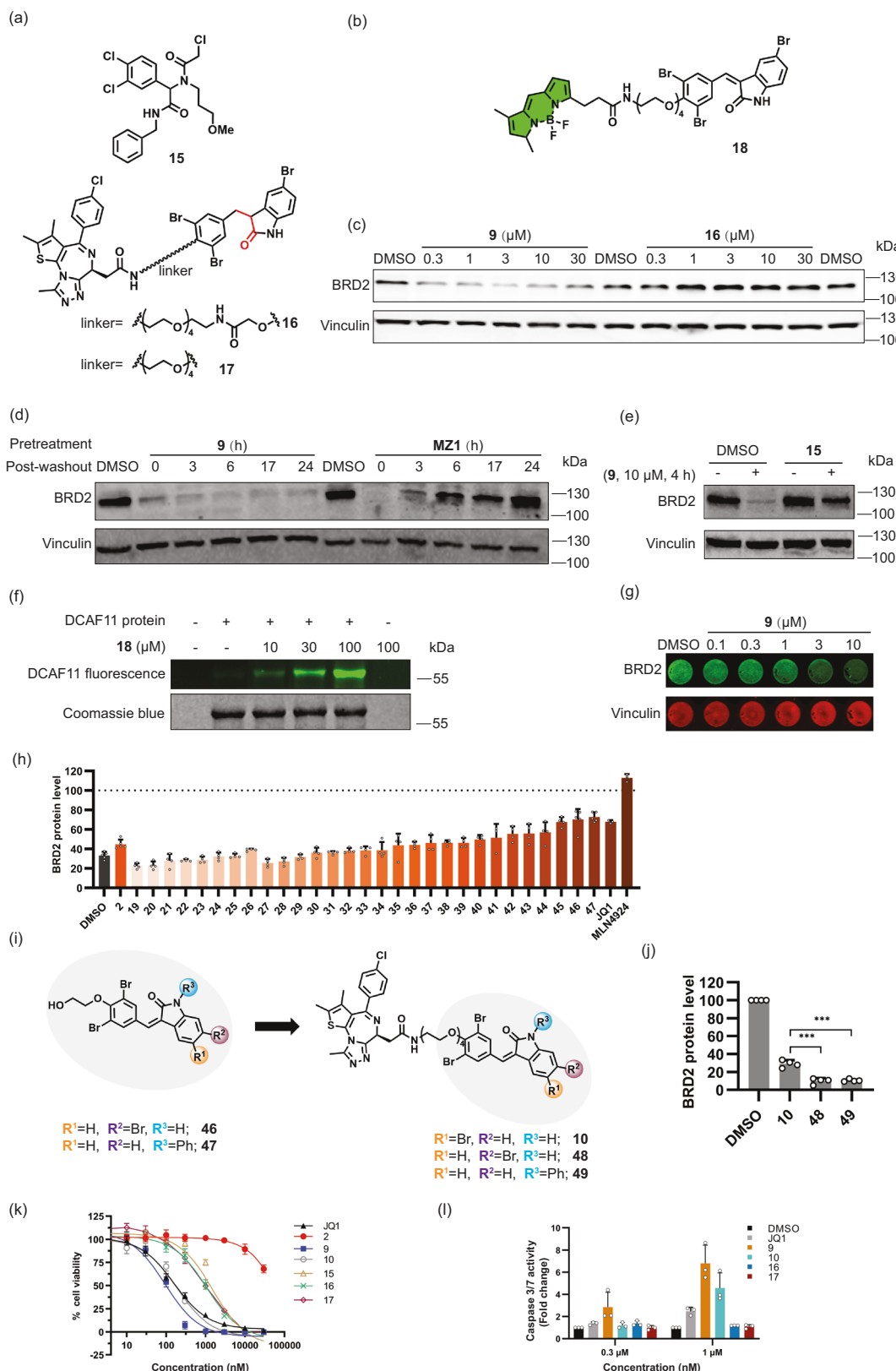

respective indolinone derivatives. A series of compounds consisting of 29 arylidene-indolinone analogs **19-47** was obtained (for structures see Supplementary Table 1) and their activity was ranked by means of the qualitative ICW assay (Fig. 4h). As a general trend, variation of the substitution pattern in the phenyl ring of the benzylidene substituent is tolerated. However, more pronounced deviations from the phenyl ring to e.g. heterocycles in the 3-arylidene substituent lead to a

decrease in activity (compare compound **2** with compounds **19-26**). Introduction of different substituents into the indolinone core is tolerated as well. In particular, a Br at the 6-position (**46**), or functionalization of the indolinone N with a phenyl group (**47**) led to increased activity (Fig. 4h, i, Supplementary Table 1 and Supplementary Fig. 7c–e). These results were confirmed by determination of the degradation activity of the corresponding bifunctional degraders **48**

**Fig. 4 | Bifunctional compounds bind covalently to DCAF11 and display anti-proliferative activity. a** Structures of **15** and bifunctional compounds **16** and **17** with saturated double bond. **b** Structure of probe **18** with BODIPY as fluorophore. **c** BRD2 levels in KBM7 cells treated with compounds **9** and **16** for 4 h. Representative result of $n = 3$. **d** BRD2 levels in Jurkat cells treated with PROTAC MZ1 or **9** (10 μM, 5 h) prior to washout and further incubation for the indicated time. Representative results of $n = 3$. **e** BRD2 levels in Hep G2 cells pretreated with 30 μM **15** for 1 h followed by 10 μM of compound **9** for another 4 h. Representative result of $n = 2$. **f** Fluorescence labelling of DCAF11. Purified DCAF11 protein was incubated with 18 for 40 min followed by analysis of in-gel fluorescence. Representative result of $n = 3$. **g** In-Cell Western for BRD2 levels in Hep G2 cells treated with **9** for 4 h. Representative results ($n = 4$ biological replicates). **h** Relative BRD2 protein levels as detected using In-Cell Western in Hep G2 cells pretreated with compounds **2, 19-47** or JQ1 (structure shown in Supplementary Table 1) at 30 μM or MLN4924 at 1 μM for 1 h, followed by **9** (10 μM, 4 h) or DMSO addition. Data are mean values ± SD ($n = 3$ for JQ1 and MLN4924, $n = 4$ for 19–45, $n = 5$ for DMSO, 2, 46–47; $n =$ biological replicates). **i** Structures of representative indolinones (**46, 47**) and corresponding bifunctional compounds (**48, 49**). **j** BRD2 levels in Jurkat cells treated with **9, 48** or **49** at 3 μM for 6 h. Data are mean values ± SD ($n = 4$ biological replicates). Statistical significance was calculated with unpaired two-tailed Student's t-tests comparing 10 to 48 or 49 ($P = 0.0002$ or 0.0003 respectively; ***$P < 0.001$). **k** Influence on cell viability. Data are mean values ± SD ($n = 3$ biological replicates). The data of GI$_{50}$ are shown in Supplementary Fig. 7h. **l** Caspase-3/7 activity for apoptosis detection. Data are mean values ± SD ($n = 3$ biological replicates).

and **49** (derived from **46** and **47** respectively; Fig. 4i, j and Supplementary Fig. 8a), and their degradation activities were also based on DCAF11 (Supplementary Fig. 8b). **46** and **47** were also connected with PDEδ ligand to generate degraders **54** and **55** (Supplementary Fig. 8c), which demonstrated comparable or better degradation activity at 10 μM than degrader **5** (Supplementary Fig. 8d).

As GW5074 targets kinases and SIRTs[13–15], we explored whether indolinone **42** modulates kinases and SIRTs as well. In a panel of 482 *wildtype* and mutant kinases, compound **42** inhibited only seven *wildtype* kinases by more than 50 % (see Supplementary Data 3). Furthermore, **42** moderately suppressed the activity of only SIRT3 (Supplementary Table 2). We then assessed the binding to LC3B protein as indolinones based on GW5074 are reported to bind to the protein. The BODIPY labelled compound **18** failed to bind to LC3B in a fluorescence polarization experiment (Supplementary Fig. 8e). In line with this result, compound **5** did not change the thermal stability of LC3B, while the known LC3B binder DC-LC3in-D5[33] increased the melting temperature of LC3 by 13 °C (Supplementary Fig. 8f).

## BET-targeting degraders are cytotoxic and induce apoptosis

In order to assess potential cytotoxic effects induced by the electrophilic indolinone moiety, BET ligand JQ1 (**8**), DCAF11 ligands **2** and **15**, active bivalent degraders **9** and **10** and non-active compounds **16** and **17** were subjected to viability assays employing Jurkat and Hep G2 cells (Fig. 4k and Supplementary Fig. 8g, h). Monovalent DCAF11 ligand **2**, which embodies the electrophilic arylidene-indolinone scaffold, was only weakly antiproliferative in Jurkat- and Hep G2 cells (GI$_{50}$ > 30 μM), whereas the covalent chloroacetamide DCAF11 ligand **15** showed higher toxicity in both cell lines, with GI$_{50}$ values of 1.7 ± 0.2 μM and 6.5 ± 1.8 μM. Active JQ1-derived bivalent degraders **9** and **10** displayed antiproliferative activity in Jurkat cells with GI$_{50}$ values of 0.09 ± 0.02 μM and 0.19 ± 0.03 μM. They are 6- to 10-fold more potent than non-electrophilic compounds **16** and **17** which display GI$_{50}$ values in the range of 1.0 to 1.2 μM. BET protein degradation is associated with induction of apoptosis[34,35]. To analyze apoptosis induction by the DCAF11-targeting degraders, Jurkat cells were treated with JQ1 or compounds **9, 10, 16** and **17** for 18 h, followed by detection of caspase-3/7 activity. In the presence of degraders **9** and **10**, caspase-3/7 activation was substantially higher than after treatment with JQ1 or non-active degraders **16** and **17**, indicating that **9** and **10** are more potent apoptosis inducers than ligand JQ1 (Fig. 4l).

## Discussion

Targeted protein degradation (TPD) by means of bifunctional degraders that direct proteins of interest to the ubiquitin-proteasome system (UPS) or to degradation by macroautophagy is a novel approach to modulate protein activity and function for chemical biology and medicinal chemistry investigations. In particular, it enables the modulation of proteins considered undruggable by established small-molecule approaches.

In an attempt to explore whether effective degradation of the lipoprotein binding chaperone PDEδ could be achieved by targeting it for degradation through macroautophagy, we designed bifunctional modalities combining a 3-arylidene-indolinone reported before to bind to the autophagy-related protein LC3B[6,8] and PDEδ-inhibitors previously developed by us[11,19]. Biological characterization of these PDEδ-degraders as well as analogous degraders targeting the BRD2/3/4 proteins or the BTK- and BLK kinases unexpectedly revealed that these bifunctional compounds induce degradation of their target proteins through the UPS and not by means of autophagy. Subsequent target identification and validation as well as mechanistic investigations showed that the CRL4$^{DCAF11}$ E3 ligase complex is responsible for proteasomal degradation, and that the 3-arylidene-indolinones covalently bind to the substrate receptor protein DCAF11.

Recently, Cravatt et al. reported that the DCAF11 protein can be targeted covalently by chloroacetamide derivatives that bind to a patch of cysteines. These electrophiles were considered valuable tool compounds for cellular investigations, however, the authors stress that the discovery of degraders incorporating more tempered electrophiles, such as α, β-unsaturated amides would be highly desirable[18]. High electrophilic reactivity may lead to undesired targeting of multiple proteins at cysteine- and lysine sites, thereby inducing off-target-mediated cytotoxicity. Electrophilic 3-arylidene-indolinones can undergo conjugate addition of thiol nucleophiles[36], and should, in principle, also be able to covalently bind to cysteines in the DCAF11 binding site. Intriguingly, the guiding compound GW5074 **1** has been extensively explored in kinase- and SIRT-inhibitor discovery[13–15], is an established research tool[16] and can be considered a close analog of the marketed kinase inhibitor sunitinib **4**. However, to the best of our knowledge, neither for sunitinib, nor for analogs like GW5074 has covalent binding to cysteines in the binding sites of their target proteins been reported. Hence, their ability to act as covalent modulators of protein activity identified here is unprecedented, and their electrophilic potency appears to be significantly lower than the reactivity observed for chloroacetamides. Therefore, the drug-like arylidene-indolinones define a DCAF11-ligand class that may enable the wider exploration of PROTACs targeting this E3 ligase in medicinal chemistry programs.

For TPD employing bifunctional PROTACs currently only a few of the more than 600 E3 ligases in mammalian cells have been investigated, and mostly the von Hippel-Lindau (VHL)- and Cereblon (CRBN) proteins have been targeted[5]. However, chronic treatment with PROTACs based on VHL and CRBN has given rise to inactivating mutations in the E3 ligases leading to resistance[29–31]. Also, induction of target degradation may vary among ligases, which may be linked to expression levels of ligases in different tissues or cell types, their particular subcellular localization, or different structural matches in the interaction interfaces formed between ligases and targets. Therefore, the discovery of new E3 ligases, which may offer novel opportunities for drug discovery, is in high demand but in short supply. The findings reported here together with the results of Cravatt et al.[18] suggest that DCAF11 may be a very suitable candidate ligase worth exploring for chemical biology and medicinal chemistry and drug discovery programs in a broader sense.

Finally, we note that the unsaturated 3-arylidene-indolinone scaffold frequently occurs in natural products such as *E-7*[12] (compound **3**, Fig. 1a). The same holds true for the core cyclic imide defining the degron recognized by Cereblon[37], which occurs for instance in natural products such as the Neolaugerines, the Wasalexins and the Costinones[12]. In addition, it has been found that electrophilic natural products such as nimbolide[38], manumycin polyketides[39] and piperlongumine[40] target E3 ligases. This insight raises the question whether nature, in a more general sense, has employed small molecule-mediated protein degradation as a means to regulate protein concentration and function, and if additional natural E3 ligase-targeting compound classes can be identified. A broader investigation of natural products[41] and natural product-inspired compound classes such as pseudo-natural products[42–44] may tell.

## Methods

### Reagents and antibodies

Following reagents were used: Chloroquine (Sigma, C6628), NH$_4$Cl (J.T.Baker, 0018), and Bafilomycin A1 (Cell Signaling Technology (CST), 54645S), Carfilzomib (Abcam, ab216469), MG-132 (CST, 2194S), MLN4924 (CST, 85923S), and JQ1 (Sigma-Aldrich, SML1524). Deltazinone is from COMAS. Compounds **18-38** were purchased from ChemDiv. Details of chemical syntheses and NMR spectrum of other compounds can be found in the Supplementary Information as Supplementary Fig. 9–20. The antibodies for BRD2 (5848 S), BRD4 (13440S), LC3B (2775S), GAPDH (2118S), DDB1 (6998S), BTK (8547S), BLK (3262 S), FIP200 (12436S), V5-Tag (13202S) and FLAG-Tag (2368S) were purchased from Cell Signaling Technology. The antibodies for BRD2 (for In-Cell Western, ab139690), BRD3 (ab50818), and RBX1 (ab133565) were purchased from Abcam. The antibody for PDEδ (PA5-22008) was purchased from Invitrogen. The antibody for Vinculin (V9131) was purchased from Sigma-Aldrich, and the antibody for DCAF11 (A15519) was purchased from ABclonal. HRP-conjugated streptavidin (Thermo Fisher, N100) was used for the visualization of biotinylated proteins. The HRP secondary antibodies (31460 for anti-rabbit and 62-6520 for anti-mouse) were purchased from Invitrogen. APC anti-CD90.1/Thy1.1 (202526, BioLegend) and Human TruStain FcX™ Fc Receptor Blocking Solution (422302, BioLegend) were used for FACS-based CRISPR/Cas9 screens.

**Cell culture.** All mammalian cells were cultured under sterile conditions in a humidified atmosphere at 5% CO$_2$ at 37 °C. Jurkat (ACC-282, DSMZ, RID: CVCL_0065) and Ramos cells (ACC-603, DSMZ, RRID: CVCL_0597) were cultured in RPMI1640 medium with 10% FBS. HEK293T (CRL-11268, ATCC, RRID: CVCL_1926), Lenti-X 293T lentiviral packaging cells (Takara, #632180), MDA-MB-231 (ACC−732, DSMZ, RRID: CVCL_0062) and Hep G2 cells (ACC-180, DSMZ, CVCL_0027) were cultured in DMEM medium with 10% FBS, 1 mM sodium pyruvate and non-essential amino acids. KBM7 (from T. Brummelkamp lab) and HAP1 cells (C631, Horizon, RRID: CVCL_Y019) were cultured in IMDM medium with 10% FBS. All cell lines were regularly tested for mycoplasma contamination and were always free of mycoplasma.

**Western Blot.** Cells were washed with PBS and lysed in lysis buffer (50 mM Tris-HCl, pH 7.4, 150 mM NaCl, 1% Triton X-100, 1% sodium deoxycholate, 0.1% SDS) with complete protease inhibitor cocktail (Roche, 11836170001) for 30 min on ice. For Ramos cells, cell pellets were lysed in lysis buffer (50 mM Tris-HCl, pH 8, 150 mM NaCl, 1% NP-40, 0.5% sodium deoxycholate) with complete protease inhibitor cocktail. After centrifugation at 16,000 x g, the supernatant was transferred into a new tube and the protein concentration was determined by DC protein assay (BIO-RAD, 5000116). Proteins were separated by SDS-PAGE and transferred to a PVDF membrane (Thermo scientific, 0.45 μm, 88518). For PDEδ, BRD2, GAPDH, BRD4, Vinculin and BTK, the membranes were blocked with Intercept® Blocking Buffer

(LI-COR Biosciences, 927-70001) at room temperature for 1 h. The primary antibodies for these proteins were diluted with Intercept® Blocking Buffer (1:500 for PDEδ, 1:3000 for Vinculin, and 1:1000 for other proteins) and incubated with the membranes overnight at 4 °C. Membranes were washed with PBS + 0.1% Tween 20 (PBS-T) five times and incubated with secondary antibodies conjugated to IRDye® Infrared Fluorescent Dyes (1:5000, LI-COR Biosciences) at room temperature for 1 h with protection from light. After washing the membrane, signals were visualized using the ChemiDoc MP Imaging System (BIO-RAD Laboratories). For detection of LC3B, BRD3, BLK, DCAF11, DDB1 RBX1, FIP200, V5- and FLAG-Tag the membranes were blocked with 5% milk in PBS-T at room temperature for 1 h. The primary antibodies for these proteins were diluted in 5% milk in PBS-T buffer (1:200 for BRD3, 1:1000 for other proteins) and incubated with the membranes overnight at 4 °C. Membranes were washed with PBS-T five times and incubated with HRP secondary antibodies (1:1500) at room temperature for 1 h. After washing the membrane, protein detection was performed using Western blotting detection reagent (Thermo Scientific, 34095) and the ChemiDoc MP Imaging System (BIO-RAD). Relative band intensities were quantified using Image Lab (BIO-RAD).

**RT-qPCR.** Jurkat cells were grown in 12-well plates and treated with compounds for the indicated time. RNA was extracted using the RNeasy Plus Mini Kit (QIAGEN, 74134), and DNA was removed by DNase digestion with RNase-free DNase Set (QIAGEN, 79256). cDNA was synthesized using the QuantiTect Reverse Transcription Kit (QIAGEN, 205313). The mRNA expression level was evaluated with the QuantiFast SYBR Green PCR Kit (Qiagen, 330502), using the CFX96 Real-Time PCR Detection System (BIO-RAD Laboratories), and relative expression levels were calculated using the $\Delta\Delta C_t$ method using GAPDH as reference[45] and analyzed in GraphPad Prism 9.0. The sequences of the primers for *PDE6D* were 5'-GCAGTCCTTGATAGAGGCAGCA-3' (forward) and 5'-GAAAAGTCTCACTCTGGATGTGC-3' (reverse). For *GAPDH*, the sequences were 5'-GTCTCCTCTGACTTCAACAGCG-3' (forward) and 5'-ACCACCCTGTTGCTGTAGCCAA-3' (reverse). For *BRD2*, the sequences were 5'-CGGCTTATGTTCTCCAACTGCTA-3' (forward), 5'-GGCAGTAGAGACTGGTAAAGGC-3' (reverse). For *BRD3*, the sequences were 5'-CCAACCATCACTGCAAACGTCAC-3' (forward), 5'-GGAGTGGTTGTGTCTGCTTTCC-3' (reverse). For *BRD4*, the sequences were 5'-CGCTATGTCACCTCCTGTTTGC-3' (forward), 5'-ACTCTGAG-GACGAGAAGCCCTT-3' (reverse). These primers were synthesized by Sigma-Aldrich.

**RNA interference.** Hep G2 cells and MDA-MB-231 cells were seeded in 6-well plates and non-targeting (NT) siRNA (Horizon Discovery Ltd, D-001810-10-05), DCAF11 siRNA (Horizon Discovery Ltd, SMARTPool, L-019051-01-0005) or RBX1 siRNA (Horizon Discovery Ltd, SMARTPool, L-004087-00-0005) were transfected into the cells with DharmaFECT 1 transfection reagent (Horizon Discovery Ltd, T-2001-02) following the manufacturer's protocol. After incubation with NT siRNA (50 nM, 72 h) or DCAF11 siRNA (50 nM, 72 h) in Hep G2 cells, or incubation with NT siRNA (30 nM, 48 h) or RBX1 siRNA (30 nM, 48 h) in MDA-MB-231 cells, 3 μM compound 9 was added for another 5 h or 4 h respectively and cells were collected and lysed as mentioned above.

**CRISPR–Cas9 knockout screens.** For pooled FACS-based CRISPR–Cas9 BRD4 protein stability screens, a CRL-focused sgRNA library[26] was lentivirally packaged using polyethylenimine (PEI MAX® MW 40,000, Polysciences) transfection of Lenti-X cells and the lentiviral pCMVR8.74 helper (Addgene plasmid # 22036) and pMD2.G envelope (Addgene plasmid # 12259; both gifts from Didier Trono) plasmids. The virus containing supernatant was cleared of cellular debris by filtration through a 0.45-μm PES filter and used to transduce KBM7 BRD4-BFP reporter cells harboring a doxycycline inducible Cas9 allele at a multiplicity of infection (MOI) of 0.05 and 1,000-fold library

representation. Library-transduced cells were selected with G418 (1 mg ml⁻¹, Gibco) for 14 days, expanded and Cas9 expression was induced with DOX (0.4 µg ml⁻¹, PanReac AppliChem).

3 days after Cas9 induction, 25 million cells per condition were treated with DMSO (1:1000), **10** (1 µM), **9** (0.5 µM) or dBET6 (10 nM, MedChem Express) for 6 hours in two biological replicates. Cells were washed with PBS, stained with Zombie NIR™ Fixable Viability Dye (1:1000, BioLegend) and APC anti-mouse CD90.1/Thy-1.1 antibody (1:400, BioLegend) in the presence of Human TruStain FcX™ Fc Receptor Blocking Solution (1:400, BioLegend), and fixed with 0.5 mL methanol-free paraformaldehyde 4% (Thermo Scientific™ Pierce™) for 30 min at 4 °C, while protected from light. Cells were washed with and stored in FACS buffer (PBS containing 5% FBS and 1 mM EDTA) at 4 °C overnight. The next day, cells were strained trough a 35 µm nylon mesh and sorted on a BD FACSAria™ Fusion (BD Biosciences) using a 85 µm nozzle. Aggregates, dead (ZombieNIR positive), Cas9-negative (GFP) and sgRNA library-negative (Thy1.1-APC) cells were excluded, and the remaining cells were sorted based on their BRD4-BFP and mCherry levels into BRD4^HIGH (10% of cells), BRD4^MID (30%) and BRD4^LOW (10%) fractions. For each sample, cells corresponding to at least 2,000-fold library representation were sorted per replicate.

Next-generation sequencing (NGS) libraries of sorted cell fractions were prepared as previously described[25]. In brief, genomic DNA was isolated by cell lysis (10 mM Tris-HCl, 150 mM NaCl, 10 mM EDTA, 0.1% SDS), proteinase K treatment (New England Biolabs) and DNAse-free RNAse digest (Thermo Fisher Scientific), followed by two rounds of phenol extraction and 2-propanol precipitation. Isolated genomic DNA was subjected to several freeze–thaw cycles before nested poly-merase chain reaction (PCR) amplification of the sgRNA cassette.

Barcoded NGS libraries for each sorted population were gener-ated using a two-step PCR protocol using AmpliTaq Gold (Invitrogen). The resulting PCR products were purified using Mag-Bind® TotalPure NGS beads (Omega Bio-tek) and amplified in a second PCR introducing the standard Illumina adapters. The final Illumina libraries were bead-purified, pooled and sequenced on a HiSeq 3500 platform (Illumina).

Screen analysis was performed as previously described[25]. Briefly, sequencing reads were quantified using the crispr-process-nf Nextflow workflow, available at https://github.com/ZuberLab/crispr-process-nf/tree/566f6d46bbcc2a3f49f51bbc96b9820f408ec4a3. For statistical analysis, we used the crispr-mageck-nf Nextflow workflow, available at https://github.com/ZuberLab/crispr-mageck-nf/tree/c75a90f67069 8bfa78bfd8be786d6e5d6d4fc455. To calculate gene-level enrichment, the sorted populations (BRD4^HIGH or BRD4^LOW) were compared to the BRD4^MID populations in MAGeCK (0.5.9)[46], using median normalized read counts. Raw sgRNA read number are presented in Supplementary Data 1, one-sided non-adjusted MAGeCK-log10 *p*-values (logP) and log2 fold-changes (lfc) are presented in Supplementary Data 2.

For representative scatter plots of hierarchical gating strategies for FACS-based CRISPR screens please see SI Figure 10.

**CRISPR-mediated DCAF11 and FIP200 knockout in KBM7 cell lines.** DCAF11- or FIP200 (*RB1CC1*)-targeting sgRNAs were lentivirally packaged using PEI transfection of Lenti-X cells. The viral super-natant was harvested after 72 h, cleared of cellular debris by filtration through a 0.45-µm PES filter and stored in aliquots at −80 °C. The appropriate amount of viral supernatant was transduced into KBM7 cells (1 × 10⁶ cells/mL) via spinfection (2000 rpm, 1 h, 37 °C) with polybrene (8 µg/mL). Two days post transduction, the cells were selected with puromycin (1 µg/mL) for a total of 5 days, after which the knockout pools were validated via western blotting.

**Flow cytometry-based BRD4 reporter assay**
Stable BRD4(S)-BFP reporter KBM7 iCas9 cell line was generated as previously described[47]. To quantify the influence of genetic perturba-tion on compound-induced BRD4(S)-BFP degradation, the reporter

cell line was transduced with lentiviral DCAF11-targeting sgRNA (pLenti-U6-sgRNA-IT-EF1αs-Thy1.1-P2A-NeoR) and/or transgene DCAF11 wild type or triple alanine mutant (C443A, C460A, C485A) expression vectors (pRRL-SFFV-3xFLAG-DCAF11-EF1αs-iRFP670) to 30-50% transduction efficiency. Cas9 expression was induced with dox-ycycline (0.4 µg/mL) for 4 days, followed by 6 hours treatment with **10** (5 µM), **9** (1 µM), dBET6 (0.5 µM) or DMSO. Cells were stained for sgRNA expression with an APC conjugated anti-mouse -CD90.1/Thy1.1 anti-body (1:400; Biolegend, #202526) and Human TruStain FcX Fc receptor blocking solution (1:400; Biolegend, #422302) in FACS buffer (PBS containing 5% FBS and 1 mM EDTA) for 5 minutes at 4 °C. Cells were subsequently washed, resuspended in FACS buffer and analyzed on LSRFortessa (BD Biosciences). Flow cytometry data was analyzed in FlowJo (v10.8.1). BFP and mCherry mean fluorescence intensity (MFI) values for were normalized by background subtraction of the respec-tive values from reporter negative KBM7 wild type cells. BRD4 abun-dance was calculated as the ratio of background subtracted BFP to mCherry MFI and is displayed normalized to DMSO-treated sgRNA/cDNA double negative cells.

**Cloning of constructs.** The *BRD2* cDNA with N-terminal FLAG tag for mammalian cell expression was obtained by inserting the *BRD2* cDNA (Horizon Discovery, MHS6278-202759982) into the pOPIN-E vector with the forward primer (5′- AGGAGATATACCATGGATT ACAAGGATGACGACGATAAGGGATCTATGCTGCAAAACGTGACTCC CC-3′) and reverse primer (5′-GTGATGGTGATGTTTTTAGCCTGA-GTCTGAATCACTGGTGTCTG-3′) using the In-Fusion HD Cloning Kit (Takara Clonetech, 638909). These primers were synthesized by Sigma-Aldrich. The BRD4-BFP reporter was engineered by inserting BRD4 short isoform (amino acids 1-719; Twist Bioscience) and mTagBFP into a modified pRRL lentiviral backbone[48]. mTagBFP2-pBAD was a gift from Michael Davidson (Addgene plasmid # 54572)[49]. sgRNA-resistant DCAF11 wild type and triple alanine mutant (C443A, C460A, C485A) cDNAs used for knockout/rescue studies were cus-tom synthesized by Twist Biosciences, coupled to a 3xFLAG-Tag and cloned into a pRRL lentiviral vector expressing iRFP670 for flow cytometry detection. DCAF11- and FIP200 (*RB1CC1*)-targeting sgRNAs were designed using the Vienna Bioactivity CRISPR score portal[50] (www.vbc-score.org) and were cloned into pLentiCRISPRv2 (Addgene #52961) or pLenti-U6-sgRNA-IT-EF1αs-Thy1.1-P2A-NeoR (gift from Johannes Zuber) following standard protocol[25,51]. The fol-lowing primer sequences were used: 5′-CACCGAGAGTTGGAGATCA-GATACC-3′ (forward), 5′-AAAC GGTATCTGATCTCCAACTCTC-3′ (reverse) for *DCAF11* KO1; 5′- CACCGGATGGACGAGAAGTACTAGG-3′ (forward), 5′-AAACCCTAGTACTTCTCGTCCATCC-3′ (reverse) for *DCAF11* KO2; 5′-CACCGTTTCTAACAGCTCTATTACG-3′ (forward), 5′-AAACCGTAATAGAGCTGTTAGAAAC-3′ (reverse) for *FIP200* KO1; 5′-CACCGACTACGATTGACACTAAAGA-3′ (forward), 5′-AAACTCTT-TAGTGTCAATCGTAGTC-3′ (reverse) for *FIP200* KO2.

**Co-immunoprecipitation.** Hep G2 cells were seeded in 6-well plates, 24 h later, cells were transiently transfected with HA-DCAF11 (Sino Biological, HG16257-NY) and FLAG-BRD2 for 48 h using the FuGENE® HD transfection reagent (Promega, E2311). After treatment with MG-132 (10 µM, 40 min) and compound **9** (10 µM, 1.5 h), cells were col-lected and lysed in lysis buffer (25 mM Tris-HCl pH 7.4, 150 mM NaCl, 10% glycerol, 1% NP-40) with complete protease inhibitor cocktail (Roche) on ice for 30 min, followed by centrifugation at 16,000 x g for 15 min collection of the supernatant. After determination of the pro-tein concentration, cell lysates were incubated with anti-FLAG mag-netic beads (30 µl / sample, Sigma, M8823) at 4 °C for 4 h and washed three times with washing buffer (0.2% NP-40, 25 mM Tris-HCl pH 7.4, 150 mM NaCl). The beads were diluted with PBS, mixed with SDS sample buffer and heated at 95 °C for 10 min, followed by western blot analysis.

**In-gel fluorescence.** Purified DCAF11 protein (1 μg, Origene, TP760719) was diluted in PBS buffer that is mixed with compound **2** or BODIPY derivative **18** at the indicated concentration at room temperature. Samples were diluted with 5 × SDS sample loading buffer and heated at 95 °C for 5 min. Proteins were separated using SDS-PAGE and were visualized using the ChemiDoc MP Imaging System (BIO-RAD) at 488 nm and the results were analyzed by Image Lab (BIORAD).

**Affinity-based enrichment (biotin-streptavidin pulldown).** Hep G2 cells (1 × 10⁶ cells/well) were seeded in 6-well plates and incubated for 24 h. On the next day, the cells were transfected with 1 μg pCMV3-N-HA-DCAF11 plasmid (Sino Biological, HG16257-NY) or 1 μg empty vector (pcDNA3.1(+), Thermo Fisher, V79020) and 2.5 μL FuGENE® HD transfection reagent (Promega, E2311) per well. The cells were incubated for 48 h prior to treatment with 100 μM compound **2** for 1 h. Afterwards, the biotinylated affinity probe **14** (10 μM) was added and the cells were incubated for 2 h. The cells were washed with ice-cold PBS, then lysed in lysis buffer (150 mM sodium chloride, 1% NP-40, 50 mM TRIS-HCl, pH 8.0 with complete protease inhibitor cocktail (Roche, 11836170001)) for 30 min on ice. After centrifugation at 16,000 x g, the supernatant was transferred into a new tube and the protein concentration was determined using the DC protein assay (BIO-RAD, 5000116).

For the affinity enrichment, 30 μL of Pierce™ Streptavidin Magnetic Beads (Thermo Fisher, 88817) were pre-equilibrated with 500 μL lysis buffer for 3 min at room temperature with overhead rotation. The supernatant was discarded and 1 mg of lysate in 500 μL lysis buffer was added to the beads and incubated for 1 h at room temperature with overhead rotation. The beads were washed twice with 500 μL washing buffer (50 mM PIPES (pH 7.4), 50 mM NaCl, 75 mM MgCl₂, 5 mM EGTA, 0.1% NP-40, 0.1% Triton X-100, 0.1% Tween20) and twice with PBS for 10 min each at room temperature with overhead rotation. Enriched proteins were eluted by boiling of the beads at 95 °C for 5 min in 20 μL of a supersaturated (25 mM) biotin solution and visualized by Western Blotting as described above.

**In-Cell Western.** Hep G2 cells were seeded in 96-well plates and incubated for 24 h. Compounds were added for the indicated time at the indicated concentration. Cells were then washed once with PBS and fixed with 4% paraformaldehyde for 30 min at room temperature. The fixed cells were permeabilized with 0.5 % Triton-X100 for 15 min followed by block with Intercept® Blocking Buffer (LI-COR Biosciences, 927-70001) for 1 h at room temperature. Primary antibodies for BRD2 (1:800) and Vinculin (1:2000) were diluted with LI-COR buffer and incubated with the cells at 4 °C overnight. Cells were washed with PBS-T, and the secondary antibodies conjugated to IRDye (1:1000, LI-COR Biosciences) were added to the cells for 1 h at room temperature. After washing the cells, the signal was recorded with the Odyssey® CLx Infrared Imaging System (LI-COR Biosciences) and the protein levels were analyzed by Image Studio (LI-COR Biosciences). BRD2 protein levels were normalized to the levels of reference protein vinculin.

**Cell viability assay.** Jurkat cells (3 × 10³) and Hep G2 cells (3.5 × 10³) were seeded in 96-well plates in 100 μL culture medium. Cells were treated with compounds dissolved in 50 μl medium at the indicated concentration for 72 h, at the same time, one plate of cells without compounds treatment was used for the cell count at time zero. After adding 40 μl of CellTiter-Glo® reagent (Promega, G7571) to each well for 10 min at room temperature, the luminescence was measured on a Spark® Multimode Microplate Reader (Tecan). The data was analyzed by the GraphPad Prism 9.0.

**Caspase-3/7 Activity Assay.** Jurkat cells (3 × 10³) were seeded in 96-well plates in 25 μl culture medium. Cells were treated with compounds dissolved in 25 μl medium at the indicated concentration for

18 h. After adding 50 μl of Caspase-Glo® 3/7 Reagent (Promega, G8091) to each well for 30 min at room temperature, the luminescence was measured on a Spark® Multimode Microplate Reader (Tecan). The data was analyzed by the GraphPad Prism 9.0.

**Purification of LC3B protein.** The LC3B open reading frame was cloned into pOPINE-NHis by the Dortmund protein Facility (DPF). pOPINE-NHis-LC3B was transformed into BL21DE3 RIL K+. Cells were cultured in TB medium supplemented with ampicillin and chloramphenicol (Amp/Cam), 1% glucose and 2 mM MgSO₄. A 10 L culture was incubated in TB/Amp with 0.01 % lactose monohydrate, 2 mM MgSO₄ for 4 h at 37 °C followed by overnight incubation at 20 °C. Cells were lysed using sonication in a buffer containing 50 mM HEPES (pH 7.5), 300 mM NaCl, 10 mM imidazole, 1 mM 2-mercaptoethanol, 1 mM PMSF and DNase. Upon centrifugation, the supernatant was applied onto Ni NTA column. The column was washed using 50 mM HEPES (pH 7.5), 300 mM NaCl, 30 mM imidazole, 1 mM 2-mercaptoethanol. The protein was eluted in 50 mM HEPES pH 7.5, 300 mM NaCl, 300 mM imidazole, 1 mM 2-mercaptoethanol prior to size exclusion chromatography using 26/60 G75 column and 25 mM HEPES pH 7.5, 150 mM NaCl and1 mM DTE.

**Fluorescence polarization.** The assay was performed in black, 384 well plates with a total volume of 20 μl per well using a buffer solution comprising 50 mM Tris, 300 mM NaCl and 1 mM 2-mercaptoethanol, pH 7.5. The BODIPY labelled compound **18** was tested at a final concentration of 100 μM, and the LC3B protein was titrated in two-fold dilution series. The plates were incubated at 20°C for 30 min, and fluorescence polarization was measured using a Tecan Spark plate reader at ex/em 502 / 511 nm.

**Thermal shift assay.** For thermal shift analysis, 15 μM of human LC3B was dispensed into 384-well plates (Hard-Shell 384-well 480 PCR plates, BioRAD #HSR4805) in buffer (50 mM Tris/HCl pH 7.4, 100 mM NaCl, 1 mM GSH and 0.02% CHAPS) with a Certus Flex liquid dispenser (Gyger). 60 μM compounds in DMSO (n4 or DMSO (0.6%) as control were added with an Echo 520 acoustic liquid handler (Labcyte). Glo-Melt dye was added (final concentration: 1/100), plates sealed with Lightcycler 480 Sealing Foils (Roche, #04 729 757 001) and mixed by short centrifugation (1 min, 60 g and room temeprature). A Lightcycler 480 II (Roche) was used for thermal shift curves with a temperature gradient from 20 to 95 °C according to the manufacturer´s protocol. Protein melting curves were analyzed, and the difference in meting temperature ($\Delta T_m$) was calculated with LightCycler Thermal Shift Analysis software (ROCHE) and visualized by Graphpad Prism (Version 9.5.1).

**SIRT activity assays.** In vitro enzymatic SIRT assays were performed by BPS Bioscience for SIRT1, SIRT2, SIRT3, SIRT5 and SIRT6 using the respective SIRT Fluorogenic Assay Kit (BPS Bioscience, catalog numbers 50081, 50087, 50088, 50085 and 50022). The compound was preincubated in for 30 min at room temperature in a mixture containing SIRT assay buffer, 5 μg BSA, SIRT enzyme. Afterwards, the enzymatic reaction was initiated by the addition of SIRT substrate to a final concentration of 10 μM. The enzymatic reaction proceeded for 30 min at 37 °C followed by the addition of 2 x SIRT Developer and incubation at room temperature for 15 min. Fluorescence intensity was measured at an ex/em 360/460 nm using a Tecan Infinite M1000 microplate reader.

**Kinase profiling.** Compound **42** was profiled in duplicate at 10 μM against wildtype and mutant 482 kinases using the SelectScreen Kinase profiling services (ThermoFisher Scientific, USA) using enzymatic assays (Z′-Lyte and Adapta technology) or binding assays (LanthaScreen Eu kinase binding assays) (see https://www.thermofisher.com/de/

de/home/products-and-services/services/custom-services/screening-and-profiling-services/selectscreen-profiling-service/selectscreen-kinase-profiling-service.html).

**Reporting summary**

Further information on research design is available in the Nature Portfolio Reporting Summary linked to this article.

## Data availability

Data in the FACS-based CRISPR screen are presented in supplementary Data 1 and Data 2. The inhibition activities of 42 against kinases generated in this study are presented in supplementary Data 3. The remaining data are available within the paper, Supplementary Information or Source Data File. Source data are provided with this paper.

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

## Acknowledgements

This work was supported by the Max Planck Society. CeMM and the Winter laboratory are supported by the Austrian Academy of Sciences. The Winter lab is further supported by funding from the European Research Council (ERC) under the European Union's Horizon 2020 research and innovation program (grant agreement 851478), as well as by funding from the Austrian Science Fund (FWF, projects P32125, P31690 and P7909). J.X. acknowledges the Alexander von Humboldt Stiftung for a post-doctoral fellowship. We are thankful to Johannes Zuber and Michaela Fellner for constructing the CRL-focused sgRNA library and sharing inducible Cas9 cell lines. We thank Zhou Zhao, Siska Führer and Dr. Malte Gersch for help in construction of plasmids. We thank Petra Janning, Aylin Binici, Christine Nowak, Jens Warmers and Sarah Seidel for discussion, technical assistance or data analysis. We thank the Dortmund Protein Facility for cloning the LC3B construct. We thank the Core Facility Flow Cytometry of the Medical University of Vienna for access to flow cytometry instruments and assistance with flow cytometric cell sorting. We thank Dr. Slava Ziegler for insightful discussions and proofreading of the manuscript.

## Author contributions

H.W. designed the project. G.X. and J.X. designed and synthesized compounds. G.X. performed most of the biological experiments and analyzed data. M.H. performed and analyzed CRISPR/Cas9 knockout screen. M.C. generated DCAF11 knockout and reconstitution KBM7 cell lines and analyzed compound activity in these cells. L.D. performed the affinity pulldown, PROTAC-related experiments, the fluorescence polarization studies and data analysis. H.I. performed computational and statistical analyses of CRISPR/Cas9 screens. P.L. performed the DSF measurements. X.C. synthesized compounds. S.R.A. performed some biological experiments. G.E.W. supervised the CRISPR/Cas9 screens and knockout experiments. G.X. analyzed the data. G.X., J.X. and H.W. wrote the manuscript. All authors discussed the results and commented on the manuscript.

## Funding

## Competing interests

G.E.W. is scientific founder and shareholder of Proxygen and Solgate. The Winter lab received research funding from Pfizer. The remaining co-authors declare no conflicts of interest to disclose.
