## [Peer Review File · Nature Communications]

Reviewers' Comments:

Reviewer #1:

Remarks to the Author:

Remarks to the Authors

The authors describe a new and interesting covalent bi-functional targeted protein degradation discovery story that identifies new more drug-like molecules to covalently target the Cullin-Ring Ligase (CRL) substrate receptor DCAF11, which opens possibilities towards advancing this ligase beyond being a tool of chemical biology and perhaps towards more therapeutic application. This is an area very relevant to those in the field. Interestingly, this work stems from the authors' desire to induce full cellular degradation of PDEdelta via use of 'Autotacs' to target PDEdelta for degradation via macroautophagy. Prior work from this group demonstrated that due to inherent regulatory features of PDEdelta, that complete degradation was not feasible via 'classical' TPD (for purposes here, 'classical' TPD is defined as induced proximity to E3 ligase complex resulting in ubiquitination and proteasomal degradation of the target). The continued generation of new chemical matter for TPD is an important contribution to the field and this particular manuscript shares important new insight about the mechanism of action of a class of compounds that opens new questions about prior research and this should be shared quickly.

What are the noteworthy results?

Molecules covalently targeting DCAF11 for TPD have been described previously (see <https://doi.org/10.1021/jacs.1c00990>). That previous work calls out specifically that improved, more drug-like molecules will be needed for advancement of this work and the authors here describe such molecules and derivatives of these molecules have been reported to be advanced into in vivo studies by others. Also noteworthy is that very closely related molecules have also been described to be degraders by others however via an alternative degradation mechanism (see DOI: 10.1038/s41586-019-1722-1). Today's work demonstrates convincingly, albeit with different compounds, that these molecules work via DCAF11, in contrast to LC3-driven macroautophagy. See more comments below on this. Additionally, the demonstration that GW5074 likely works here via covalent mechanism is interesting and this reactivity has not been demonstrated previously – this class of compounds is structurally related to others that have been made into therapeutics thus opening the possibility that the authors have discovered a novel discrete chemical class that could be used as an anchor for many future TPD efforts.

Will the work be of significance to the field and related fields?

Yes. Expansion of examples of novel E3 ligases are used for targeted protein degradation are important. This work also opens new questions about the mechanism of Autotacs/macroautophagy though not really addressed directly in the manuscript.

Does the work support the claims? Is additional evidence needed?

Yes. Experimentals are solid but I would suggest some additional data to make this 100%.

1. The author's labs are familiar with creating KO lines (e.g. access to DCAF11 KO HAP1 clones, near haploid KBM7 cells, etc). To fully demonstrate that macroautophagy is not responsible for activity, could they test their molecules for degradation of PDEdelta or BRD4 or BTK in an appropriate cell line that is autophagy-defective e.g. ATG5 KO cells. Chemical suppression of autophagy is not an ideal control when there is not positive control for functioning macroautophagy mechanism. What if ATG5 KO clones do not work? I do believe the fundamental conclusions of the work reported here but if macro-autophagy deficient cells also inhibit the DCAF11 mechanism, this would be an important previously undiscovered biological connection important to TPD field.

2. To prove the covalent aspect of the mechanism, the authors perform time-dependence experiments, competition binding experiments, and wash-out experiments – all classical methods. All the data supports their conclusions however it is recommended that the authors demonstrate direct binding via in-tact MS, perhaps using a tagged DCAF11. Alternatively, based on prior work of DCAF11, the authors could make targeted mutants of DCAF11 (or compound mutants) to evaluate which cysteine (if any) is essential for the mechanism of their observations – these are gold-standard methods.

Are there any flaws in the data analysis, interpretation and conclusions? Do these prohibit publication or require revision?

No flaws that prohibit publication.

Is the methodology sound? Does it meet the expected standards in your field?

Methods are standard for those in this area of research. In the case of covalency, making cysteine-deleted DCAF11 would be a standard experiment to demonstrate dependence on target and on covalent binding. This is suggested above.

Is there enough detail provided in the methods for the work to be reproduced?

Yes

Additional comments:

The original key scientific question is still not resolved – is complete degradation of PDEdelta sufficient to regulate trafficking of essential lipidated proteins such as RAS or to have an extreme effect on cholesterol biosynthesis/metabolites? Still an important question. So what happens when tools derived from 45 or 46 are coupled with PDE6delta molecules? Does this fully degrade the target? I do not think this detracts from paper but some readers may ask themselves this question.

Several of the figures could be condensed further, some moved to supplemental section which would also help simplify some of the text. I leave it to authors and editors to determine if this is helpful.

In looking into the autotac literature to help with this review, it seems to me that several control experiments are missing from those manuscripts which leaves open the possibility that those reported molecules actually work via DCAF11. I hope that these authors are also working towards developing an enhanced understanding of their molecules relative to those reported to be macroautophagy-degraders. I see this as outside the scope of this initial work but an important thread of research that this team is ideally situated to pursue.

Is GW5074 inspired from a natural product? Not relevant to the publication but the Natural Product-inspired is used in the title of the manuscript.

Supplemental info comments

1. All chemicals are well characterized and synthetic methodology is clear.
2. Full images of western blots and other gels are present and helpful.
3. Table of indolinone derivatives was very helpful to demonstrate that a lot of work was done on this targeting element.

Reviewer #2:

Remarks to the Author:

This paper argues that the BRD4 degrader reported in Chemical Communications by J. Pei et al. (cited in ref. 6 of this manuscript) does not rely on the autophagy mechanism as originally proposed, but on degradation by the ubiquitin-proteasome system. The novelty in terms of compound design is limited, as the compounds used are similar to those in the previous study by Pei et al., albeit with the inclusion of results from structure-activity relationship studies. The major progress is in the analysis of the mechanism of action.

In the previous study by Pei et al. they showed by SPR that GW5074 interacts with the autophagy-associated protein LC3. They also concluded that the target degradation by BRD4 degrader (0.5 μ M) is inhibited by the lysosomal inhibitor CQ (1 mM), which they attributed to the autophagy mechanism. They also show that cell death of MDA-MB-231 cells induced by degrader is inhibited by 3-MA.

In contrast, this manuscript by Xue et al. shows that degradation by degraders is not disturbed by CQ (50 μ M). Using the same inhibitors, the authors reach a different conclusion than the previous

study: they show that lysosomal inhibitors other than CQ, such as Bafilomycin and ammonium chloride, are also used and do not interfere with the degradation effect. As in previous studies, MDA-MB-231 cells are used, but Jurkat, HEK293, HAP1, and Hep G2 cells are not used in previous studies.

Indirect proof using such lysosomal inhibitors is insufficient to determine whether the autophagy mechanism is dependent or not. The use of cells lacking key autophagy genes is most appropriate and should be considered in the revision.

The most appropriate cells to determine whether the autophagy mechanism is dependent or not are those lacking key autophagy genes, such as FIP200 KO cells, which lack autophagy initiation factors, but Atg5 cells, which work at a later stage of the autophagy process, are also versatile. In this paper, the authors used HAP1 cells lacking LC3B, which had been proposed as a binding partner for degraders (Fig. 1e,f). Indeed, degradation of the target (PDE δ) is also observed in LC3B KO cells, but the degradation efficiency is low, and it cannot be concluded from this figure alone that LC3B is not involved in degradation. Note that this data is for n=1 and no significant difference in target degradation was tested.

The authors identified DCAF11, a ligase involved in target degradation, inferred its involvement from CRISPR screening, and confirmed the binding of DCAF11 to a degradation tag (indolinone) by labeling the DCAF11 band with the addition of fluorescent indolinone (Fig. 4c,d). The results of this experiment indicate that the fluorescent probe 17 binds to DCAF11. Caution must be exercised, however, in interpreting the results. In these experiments, covalent probes were added to purified DCAF11 protein, and other proteins may be labeled under the same conditions if they contain cysteine residues. I request that a separate experiment be conducted in which probe 17 is administered to cultured cells.

Given the structural similarity between indolinone, such as GW5074, and Sunitinib (4), the readers are interested to see if fluorescent labeling with probe 17 can also be introduced to kinases and SIRT. It is also necessary to demonstrate whether labeling can be introduced for LC3. It should also be clarified whether indolinone is promiscuous and reactive in the application of degraders containing indolinone tags as PROTACs.

While this paper is interesting because it prompts a reconsideration of the mechanism of action of the degrader reported by Pei et al. it is unlikely to be of interest to readers of Nature Communications solely based on its current content.

To increase reader interest in this manuscript, it would be desirable to additionally examine its relationship to another prior paper, called ATTEC.

Currently, very limited number of autophagy-based degraders, in order of report, AUTACs (citation of this paper is missing in this manuscript. Please cite it.), ATTECs, and AUTOTACs.

ATTEC is not a bifunctional degrader as discussed in this study, but a molecular glue that brings mutant HTT and LC3 into close proximity and suppresses mHTT levels by an autophagy mechanism. AN1 is indolinone. It would be worthwhile to investigate whether the degradation of mHTT by AN1 is induced by DCAF11 in the revised manuscript.

Reviewer #3:

Remarks to the Author:

The manuscript by Xue et al. has defined a new drug-like DCAF11 ligand class that has the potential to enable the exploration of DCAF11 in chemical biology and medicinal chemistry programs. The manuscript is well written, and the experiments are well done and convincingly demonstrate that the molecule is a ligand for DCAF11 and has the ability to bifunctionally trigger degradation. I recommend the manuscript for publication if the authors can adequately address my comment.

Comments:

The one important question that has not been answered about these drug-like, natural product-inspired arylidene indolinones, which represent a novel DCAF11 ligand class with balanced electrophilic reactivity, is how selective they are as a ligand for DCAF11 only. Does this ligand act

on other enzymes due to its structural similarity to GW5074. This is important because the structurally close analog GW5074 has been explored in the discovery of kinase and SIRT inhibitors. To address this, I would appreciate a traditional selectivity panel of kinases and SIRTs. This could be also in addition be addressed with target engagement assays such as CETSA or TPP (PMID: 30519941).

The selectivity of the DCAF11 ligand is important since you want to create a PROTAC that only goes through the intended E3 and does as little as possible in a cell to be able to rule out undesired targets to minimize unwanted effects or have a false interpretation of data.

Reviewer #1 (Remarks to the Author):

Remarks to the Authors

The authors describe a new and interesting covalent bi-functional targeted protein degradation discovery story that identifies new more drug-like molecules to covalently target the Cullin-Ring Ligase (CRL) substrate receptor DCAF11, which opens possibilities towards advancing this ligase beyond being a tool of chemical biology and perhaps towards more therapeutic application. This is an area very relevant to those in the field. Interestingly, this work stems from the authors' desire to induce full cellular degradation of PDEdelta via use of 'Autotacs' to target PDEdelta for degradation via macroautophagy. Prior work from this group demonstrated that due to inherent regulatory features of PDE6delta, that complete degradation was not feasible via 'classical' TPD (for purposes here, 'classical' TPD is defined as induced proximity to E3 ligase complex resulting in ubiquitination and proteasomal degradation of the target). The continued generation of new chemical matter for TPD is an important contribution to the field and this particular manuscript shares important new insight about the mechanism of action of a class of compounds that opens new questions about prior research and this should be shared quickly.

What are the noteworthy results?

Molecules covalently targeting DCAF11 for TPD have been described previously (see <https://doi.org/10.1021/jacs.1c00990>). That previous work calls out specifically that improved, more drug-like molecules will be needed for advancement of this work and the authors here describe such molecules and derivatives of these molecules have been reported to be advanced into in vivo studies by others. Also noteworthy is that very closely related molecules have also been described to be degraders by others however via an alternative degradation mechanism (see DOI: 10.1038/s41586-019-1722-1). Today's work demonstrates convincingly, albeit with different compounds, that these molecules work via DCAF11, in contrast to LC3-driven macroautophagy. See more comments below on this. Additionally, the demonstration that GW5074 likely works here via covalent mechanism is interesting and this reactivity has not been demonstrated previously – this class of compounds is structurally related to others that have been made into therapeutics thus opening the possibility that the authors have discovered a novel discrete chemical class that could be used as an anchor for many future TPD efforts.

Will the work be of significance to the field and related fields?

Yes. Expansion of examples of novel E3 ligases are used for targeted protein degradation are important. This work also opens new questions about the mechanism of Autotacs/macroautophagy though not really addressed directly in the manuscript.

Does the work support the claims? Is additional evidence needed?

Yes. Experimentals are solid but I would suggest some additional data to make this 100%.

1. The author's labs are familiar with creating KO lines (e.g. access to DCAF11 KO HAP1 clones, near haploid KBM7 cells, etc). To fully demonstrate that macroautophagy is not responsible for activity, could they test their molecules for degradation of PDEdelta or BRD4 or BTK in an appropriate cell line

that is autophagy-defective e.g. ATG5 KO cells. Chemical suppression of autophagy is not an ideal control when there is not positive control for functioning macroautophagy mechanism. What if ATG5 KO clones do not work? I do believe the fundamental conclusions of the work reported here but if macro-autophagy deficient cells also inhibit the DCAF11 mechanism, this would be an important and previously undiscovered biological connection important to TPD field.

*We thank Reviewer 1 for this excellent suggestion, which we have followed up experimentally as described below. Noteworthy, a similar suggestion was raised by Reviewer 2. To this goal, we have generated autophagy-deficient KBM7 cells by knocking out FIP200 with two independent sgRNAs. Both sgRNAs generated efficient knockout pools as assessed by immunoblotting (see **new Supplementary Figure 2l**). Next, we assessed the efficacy of both disclosed BET degraders **9** and **10** in inducing BRD4 and BRD3 degradation in KBM7 wildtype cells compared to both of the FIP200 knockouts via immunoblotting (see **new Supplementary Figure 2m**). In line with the proposed mechanism of action as DCAF11-dependent PROTACs, FIP200 knockout was inconsequential for the activity of either **9** or **10**, while DCAF11 knockout prevented degradation of BRD3 and BRD4.*

New Supplementary Figure 2l.

New Supplementary Figure 2m.

New Supplementary Fig. 2l: FIP200 protein levels in wild type (WT) and FIP200 knockout (KO1 and KO2) KBM7 cells. **New Supplementary Fig. 2m:** BRD3 and BRD4 protein levels in WT KBM7 cells, two independent FIP200 knockouts and DCAF11 knockout KBM7 cells treated with compound **9** (1 μM, 6 h) or **10** (10 μM, 6h). New Supplementary Fig. 2h is representative for 2 independent experiments.

2. To prove the covalent aspect of the mechanism, the authors perform time-dependence experiments, competition binding experiments, and wash-out experiments – all classical methods. All the data supports their conclusions however it is recommended that the authors demonstrate direct binding via in-tact MS, perhaps using a tagged DCAF11. Alternatively, based on prior work of DCAF11, the authors could make targeted mutants of DCAF11 (or compound mutants) to evaluate which cysteine (if any) is essential for the mechanism of their observations – these are gold-standard methods.

*To address this important suggestion of Reviewer 1 we have decided to pursue the proposed genetic experiment. Prior work from Cravatt and colleagues (Zhang, JACS 2021) has identified a reactive cysteine hotspot in DCAF11 that is composed of three residues (C443, C460 and C485) and engaged by the reported chloroacetamide warhead (compound **15**). Since we could show that cellular treatment with **15** rescues target (BRD2) degradation elicited by our indolinone-based degrader (compound **9**, see **Figure 4e**), we hypothesized that the indolinone warhead engages the same site on DCAF11. To validate this hypothesis, we employed a rescue system where DCAF11 knockout cells expressing a fluorescent BRD4 stability reporter are stably reconstituted with sgRNA-resistant DCAF11 wild type cDNA, or with the corresponding triple Cys-Ala mutant (C443A, C460A, C485A). In support of our model, we observed that the degradation-resistant DCAF11 KO cells could be re-sensitized to both BRD4 degraders (**9** and **10**) when DCAF11 wildtype cDNA is introduced. In contrast, when DCAF11 KO cells were rescued with*

*DCAF11 triple Cys-Ala mutant cDNA, cellular BRD4 levels remained refractory to both degraders, even though both DCAF11 variants were expressed to comparable levels (new **Supplementary Figures 4g and 4h**).*

New Supplementary Figure 4g (left panel): DCAF11 knockout/rescue. KBM7 iCas9 BRD4(S)-BFP reporter cells were transduced with lentivirus expressing DCAF11-targeting sgRNA as well as two different sgRNA-resistant DCAF11 expression vectors (WT DCAF11 or triple Cys-Ala mutant). After 4 days of dox-induced Cas9 expression, cells were treated with DMSO, 9 (1 μM), 10 (5 μM) or dBET6 (0.5 μM) for 6 h and BRD4-BFP levels analysed via flow cytometry. A representative of 2 independent experiments is shown.

New Supplementary Figure 4h (right panel): Expression levels of V5-tagged BRD4(S)-BFP reporter and FLAG-tagged wild type (WT) and triple Cys-Ala mutant DCAF11 in KBM7 iCas9 cells.

*Furthermore, we performed affinity-based chemical proteomics (pull-down) using a biotinylated probe (compound **14**) of indolinone **2** to detect binding to DCAF11 in cell lysates. Compound **14** enriched DCAF11 and enrichment was reduced in the presence of free compound **2**, thus confirming targeting of DCAF11 by compound **2**.*

New Figure 3k-l: Affinity-based pull down assay. Hep G2 cells overexpressing HA-DCAF11 proteins were pretreated with or without 100 μM **2** for 1 h followed by 10 μM of compound **14** for another 2 h. Affinity-based enrichment by pull down was performed and DCAF11 protein levels were analyzed by WB.

Are there any flaws in the data analysis, interpretation and conclusions? Do these prohibit publication or require revision?

No flaws that prohibit publication.

Is the methodology sound? Does it meet the expected standards in your field? Methods are standard for those in this area of research. In the case of covalency, making cysteine-deleted DCAF11 would be a standard experiment to demonstrate dependence on target and on covalent binding. This is suggested above.

Is there enough detail provided in the methods for the work to be reproduced?

Yes

Additional comments:

The original key scientific question is still not resolved – is complete degradation of PDE δ sufficient to regulate trafficking of essential lipidated proteins such as RAS or to have an extreme effect on cholesterol biosynthesis/metabolites? Still an important question. So what happens when tools derived from 45 or 46 are coupled with PDE δ molecules? Does this fully degrade the target? I do not think this detracts from paper but some readers may ask themselves this question.

To address these questions, we connected compounds 46 or 47, explicitly mentioned by the reviewer, to a PDE δ ligand by means of a PEG linker, and generated compounds 54 and 55. Although the new compounds demonstrated comparable or better degradation activity than degrader 5, they did not induce complete degradation of PDE δ .

New Supplementary Figure 7c and 7d: c. Structures of bifunctional compounds 54 and 55. d. Dose-dependent degradation of PDE δ in Jurkat cells treated with compounds 54 and 55 (6 h). Representative result of $n = 3$ is shown.

Several of the figures could be condensed further, some moved to supplemental section which would also help simplify some of the text. I leave it to authors and editors to determine if this is helpful.

We thank Reviewer 1 for this suggestion and are happy to make further adjustments based on possible input from the editorial team at Nature Communications.

In looking into the autotac literature to help with this review, it seems to me that several control experiments are missing from those manuscripts which leaves open the possibility that those reported molecules actually work via DCAF11. I hope that these authors are also working towards developing an enhanced understanding of their molecules relative to those reported to be macroautophagy-degraders. I see this as outside the scope of this initial work but an important thread of research that this team is ideally situated to pursue.

We agree with Reviewer 1 that there is a possibility that also other degraders that are reported to depend on autophagy might have alternative mechanisms. We share the reviewer's view that such investigations are clearly outside the scope of the current study and need to be addressed in detail in future publications.

Is GW5074 inspired from a natural product? Not relevant to the publication but the Natural Product-inspired is used in the title of the manuscript.

Yes, the structure is inspired by natural products. Ref. 12 in the Introduction (Millemaggi, A. & Taylor, R. J. K. 3-Alkenyl-oxindoles: natural products, pharmaceuticals, and recent synthetic advances in tandem/telescoped approaches. Eur. J. Org. Chem. 2010, 4527-4547) is a review that summarizes knowledge about these natural products.

Supplemental info comments

1. All chemicals are well characterized and synthetic methodology is clear.
2. Full images of western blots and other gels are present and helpful.
3. Table of indolinone derivatives was very helpful to demonstrate that a lot of work was done on this targeting element.

Reviewer #2 (Remarks to the Author):

This paper argues that the BRD4 degrader reported in Chemical Communications by J. Pei et al. (cited in ref. 6 of this manuscript) does not rely on the autophagy mechanism as originally proposed, but on degradation by the ubiquitin-proteasome system. The novelty in terms of compound design is limited, as the compounds used are similar to those in the previous study by Pei et al., albeit with the inclusion of results from structure-activity relationship studies. The major progress is in the analysis of the mechanism of action.

In the previous study by Pei et al. they showed by SPR that GW5074 interacts with the autophagy-associated protein LC3. They also concluded that the target degradation by BRD4 degrader (0.5 μ M) is inhibited by the lysosomal inhibitor CQ (1 mM), which they attributed to the autophagy mechanism. They also show that cell death of MDA-MB-231 cells induced by degrader is inhibited by 3-MA. In contrast, this manuscript by Xue et al. shows that degradation by degraders is not disturbed by CQ (50 μ M). Using the same inhibitors, the authors reach a different conclusion than the previous study: they show that lysosomal inhibitors other than CQ, such as Bafilomycin and ammonium chloride, are also used and do not interfere with the degradation effect. As in previous studies, MDA-MB-231 cells are used, but Jurkat, HEK293, HAP1, and Hep G2 cells are not used in previous studies.

The reviewer is correct, indeed, we use additional cell lines and, therefore, our data rest on a broader basis than previous work.

Indirect proof using such lysosomal inhibitors is insufficient to determine whether the autophagy mechanism is dependent or not. The use of cells lacking key autophagy genes is most appropriate and should be considered in the revision. The most appropriate cells to determine whether the autophagy mechanism is dependent or not are those lacking key autophagy genes, such as FIP200 KO cells, which lack autophagy initiation factors, but Atg5 cells, which work at a later stage of the autophagy process, are also versatile.

*We agree with this comment raised by Reviewer 2 (and the largely overlapping point raised by Reviewer 1 above). To address these concerns, we have generated autophagy-deficient KBM7 cells by knocking out FIP200 with two independent sgRNAs. Both sgRNAs generated efficient knockout pools as assessed by immunoblotting (see **new Supplementary Figure 3e**). Next, we assessed the efficacy of both disclosed BET (BRD4-) degraders **9** and **10** in inducing BRD4 and BRD3 degradation in KBM7 wildtype cells compared to both of the FIP200 knockouts via immunoblotting (see **new Supplementary Figure***

3f). *In line with the proposed mechanism of action as DCAF11-dependent PROTACs, FIP200 knockout was inconsequential for the activity of either 9 or 10 in degrading BRD3 and BRD4.*

New Supplementary Fig. 3e (left panel): FIP200 protein levels in wild type (WT) and FIP200 knockout (KO1 and KO2) KBM7 cells.

New Supplementary Fig. 3f (right panel): BRD3 and BRD4 protein levels in WT KBM7 cells, two independent FIP200 knockouts and DCAF11 knockout KBM7 cells treated with compound **9** (1 μ M, 6 h) or **10** (10 μ M, 6h). New Supplementary Fig. 2h is representative for 2 independent experiments.

In this paper, the authors used HAP1 cells lacking LC3B, which had been proposed as a binding partner for degraders (Fig. 1e,f). Indeed, degradation of the target (PDE δ) is also observed in LC3B KO cells, but the degradation efficiency is low, and it cannot be concluded from this figure alone that LC3B is not involved in degradation. Note that this data is for n=1 and no significant difference in target degradation was tested.

We are grateful to the reviewer for spotting that we gave the wrong value for “n” in these experiments. Actually, n equals 3 in the work shown, and we just show one representative example. We corrected the manuscript accordingly with n=3.

Concerning the degradation efficiency, we respectfully disagree with the reviewer. In our opinion, the Figure clearly shows that degradation occurs. In addition, we now provide direct evidence (see below) that the compounds do not bind LC3, and the FIP200 knock out data in addition strengthen our findings. We hope the reviewer agrees that these cumulated data are convincing.

The authors identified DCAF11, a ligase involved in target degradation, inferred its involvement from CRISPR screening, and confirmed the binding of DCAF11 to a degradation tag (indolinone) by labeling the DCAF11 band with the addition of fluorescent indolinone (Fig. 4c,d). The results of this experiment indicate that the fluorescent probe 17 binds to DCAF11. Caution must be exercised, however, in interpreting the results. In these experiments, covalent probes were added to purified DCAF11 protein, and other proteins may be labeled under the same conditions if they contain cysteine residues. I request that a separate experiment be conducted in which probe 17 is administered to cultured cells.

As requested by the reviewer, we performed affinity-based chemical proteomics (pulldown) after treatment of HepG2 cells with a biotinylated probe (compound 14) of indolinone 2 to detect binding to DCAF11. Compound 14 enriched DCAF11 and enrichment was reduced in the presence of free compound 2, thus confirming targeting of DCAF11 by compound 2.

(k)

(l)

New Figure 3k-l: Affinity-based pull down assay. Hep G2 cells overexpressing HA-DCAF11 proteins were pretreated with or without 100 μM **2** for 1 h followed by 10 μM of compound **14** for another 2 h. Affinity-based enrichment by pull down was performed and DCAF11 protein levels were analyzed by WB.

Given the structural similarity between indolinone, such as GW5074, and Sunitinib (**4**), the readers are interested to see if fluorescent labeling with probe **17** can also be introduced to kinases and SIRT.

*As requested by the reviewer, we tested the activity of five SIRTs in the presence of indolinone **42**. Only moderate inhibition of SIRT3 was detected, whereas SIRT1,2,5 and 6 were not inhibited (see new Table S5).*

Supplementary Table S5. SIRT activity in presence of 3 μM compound **42**.

Protein	Activity [%] (N=1)	Activity [%] (N=2)	Mean activity [%]
SIRT1	93	97	95
SIRT2	104	111	107.5
SIRT3	67	67	67
SIRT5	95	98	96.5
SIRT6	87	90	88.5

It is also necessary to demonstrate whether labeling can be introduced for LC3.

*We employed the fluorescent probe **18** to explore binding to LC3 using fluorescence polarization. However, no binding of the probe to the protein was detected. We also used the orthogonal thermal shift assay and found that compound **5** did not impact the thermal stability of the LC3B protein, whereas the known LC3B binding compound DC-LC3in-D5 causes thermal stabilization of LC3B with a shift in melting temperature of 13°C (see new Figure S7e and f)*

(e)

(f)

New Figure S7e-f. Potential binding of probe **18** to LC3B was assessed by means of fluorescence polarization. **f.** Potential binding of compound **5** to LC3B as determined using DSF and Glo-Melt. Compound DC-LC3in-D5 was used a control. Melting temperature shift (ΔT_m) DC-LC3in-D5 = 13.04 °C.

It should also be clarified whether indolinone is promiscuous and reactive in the application of degraders containing indolinone tags as PROTACs.

While we formally can't exclude that the employed indolinone ligand doesn't also engage other E3 ligases and/or other effectors of the ubiquitin proteasome system, our unbiased functional genomics experiment clearly indicates the CRL4^{DCAF11} ligase as the sole, non-redundant functional effector that drives the target degradation that is induced by the degraders characterized in this manuscript.

While this paper is interesting because it prompts a reconsideration of the mechanism of action of the degrader reported by Pei et al. it is unlikely to be of interest to readers of Nature Communications solely based on its current content. To increase reader interest in this manuscript, it would be desirable to additionally examine its relationship to another prior paper, called ATTEC.

Currently, very limited number of autophagy-based degraders, in order of report, AUTACs (citation of this paper is missing in this manuscript. Please cite it.), ATTECs, and AUTOTACs. ATTEC is not a bifunctional degrader as discussed in this study, but a molecular glue that brings mutant HTT and LC3 into close proximity and suppresses mHTT levels by an autophagy mechanism. AN1 is indolinone. It would be worthwhile to investigate whether the degradation of mHTT by AN1 is induced by DCAF11 in the revised manuscript.

We respectfully disagree with the reviewers comment. The mHTT degraders have been described as molecular glues, and our compounds can not share this mode of action. They also do not bind to LC3, as is now additionally confirmed. Furthermore, the paper reporting AN1 employs primary patient material to which we do not have access.

The purpose of our paper is not to validate or devalidate the work of others. We have focused on the identification of the ligase that is responsible for degradation initiated by the PROTACs we developed. Full characterization of one compound class is a task that can be met in one paper. Extensive investigation of additional compound classes should be left to subsequent work.

Reviewer #3 (Remarks to the Author):

The manuscript by Xue et al. has defined a new drug-like DCAF11 ligand class that has the potential to enable the exploration of DCAF11 in chemical biology and medicinal chemistry programs. The manuscript is well written, and the experiments are well done and convincingly demonstrate that the molecule is a ligand for DCAF11 and has the ability to bifunctionally trigger degradation. I recommend the manuscript for publication if the authors can adequately address my comment.

Comments:

The one important question that has not been answered about these drug-like, natural product-inspired arylidene indolinones, which represent a novel DCAF11 ligand class with balanced electrophilic reactivity, is how selective they are as a ligand for DCAF11 only. Does this ligand act on other enzymes due to its structural similarity to GW5074. This is important because the structurally close analog GW5074 has been explored in the discovery of kinase and SIRT inhibitors. To address this, I would appreciate a traditional selectivity panel of kinases and SIRTs. This could be also in addition

be addressed with target engagement assays such as CETSA or TPP (PMID: 30519941). The selectivity of the DCAF11 ligand is importance since you want create a PROTAC that only goes through the intended E3 and do as little as possible in a cell to be able to rule out undesired targets to minimize unwanted effects or have an false interpretation of data.

As requested by the reviewer, we tested the activity of five SIRT in the presence of indolinone 42. This compound was chosen to guarantee relevance of the structure, because it contains the first part of the linker used in the other conjugates. Only moderate inhibition of SIRT3 was detected, whereas SIRT1,2,5 and 6 were not inhibited (see new Table S5).

Table S5. SIRT activity in presence of 3 μ M compound 42.

Protein	Activity [%] (N=1)	Activity [%] (N=2)	Mean activity [%]
SIRT1	93	97	95
SIRT2	104	111	107.5
SIRT3	67	67	67
SIRT5	95	98	96.5
SIRT6	87	90	88.5

Moreover, compound 42 was subjected to a panel of 482 wildtype and mutant kinases. Compound 42 inhibited only seven wt kinases by more than 50% (see new Table S4)

	A	E	F
1	Kinase	Method	% Inhibition
2	LRRK2 G2019S FL	Adapta	74
3	FLT3	ZLYTE	71
4	LRRK2 G2019S	Adapta	64
5	LRRK2 I2020T	Adapta	60
6	FLT3 ITD	LanthaScreen Binding	59
7	PI4KB (PI4K beta)	Adapta	57
8	LRRK2 FL	Adapta	56
9	MYLK4	LanthaScreen Binding	54
10	TNIK	LanthaScreen Binding	54
11	CAMK1 (CaMK1)	Adapta	53
12	GSG2 (Haspin)	Adapta	51
13	BRAF V599E	ZLYTE	48
14	RPS6KA1 (RSK1)	ZLYTE	46
15	LRRK2	Adapta	45
16	LRRK2 R1441C	Adapta	44
17	KIT V559D	ZLYTE	42
18	MINK1	ZLYTE	42

Reviewers' Comments:

Reviewer #1:

Remarks to the Author:

The authors have appropriately and adequately addresses points raised in the first round of review. Specifically demonstrating via both genetic and pharmacologic means that DCAF11 is necessary and sufficient for degradation in the context of their experiments, thus demonstrating their molecule working via this mechanism and not via autophagic mechanisms. Its a shame that we have not been able to fully deplete PDEdelta to get a full handle on its role in trafficking of prenylated proteins (e.g. such as RAS family) but the new tools described do enable greatly increased depletion and may prove helpful to future research.

Thank you to the authors for also including the molecular characterization of the new molecules synthesized in their follow-up experiments (e.g. cmpds 14, 54. 55) and the details of the additional experiments in SI.

I support publication of this manuscript.

Reviewer #2:

Remarks to the Author:

The authors have done an excellent job responding to the concerns previously raised by reviewers.